# HA-PAT: Hierarchically-Adaptive Pruning-Aware Tuning for Large Language Models

## Abstract

The enormous size of large language models (LLMs) limits their deployment and application. Some research utilizes structural pruning to alleviate this by removing redundant weights in a hardware-agnostic manner. However, existing methods tend to apply a uniform pruning strategy across all layers, ignoring the layer-wise functional diversity and risking the removal of essential model components. To tackle this challenge, we propose a Hierarchically-Adaptive Pruning-Aware Tuning (HA-PAT) method. Based on the pruning-aware tuning framework, HA-PAT employs Hierarchical Pruning Ratio Scheduling (HPRS) to derive optimal layer-wise sparsity guided by each layer's unique functionality. It preserves the general linguistic functions of shallow layers, while aggressively pruning the deeper layers that primarily encode task-specific features. To better preserve model performance, HA-PAT introduces a magnitude vector into the compensation mechanism, enabling the reconstruction of pruned weights based on a broader information space. Experimental results show that our method consistently outperforms the baseline both in average accuracy and inference efficiency. On LLaMA2-13B with 25% pruning ratio, our approach surpasses the PAT baseline by 4.01% in average accuracy across 14 benchmarks, along with a 30% inference speedup. Further experiments on downstream tasks indicate that HA-PAT better preserves the pre-trained language understanding capabilities.

## 1 Introduction

Large language models (LLMs) (Brown et al., 2020; Touvron et al., 2023b; Achiam et al., 2023) have achieved revolutionary breakthroughs in numerous fields of natural language processing. Their enormous performance is primarily attributed to their massive parameter scale (Kaplan et al., 2020). The continued growth in the size of LLMs has introduced significant computational and memory costs that severely limit their practical deployment in resource-constrained environments (Patterson et al., 2021; Narayanan et al., 2021; Wan et al., 2023), making model compression research crucial for reducing these overheads.

To address this challenge, various model compression techniques have been proposed, including quantization (Dettmers et al., 2022; Liu et al., 2024b), knowledge distillation (Sanh, 2019; Tunstall et al., 2023), and model pruning (Goyal et al., 2020; Frantar & Alistarh, 2023). Quantization reduces model size by lowering the numerical precision of parameters (e.g., from FP16 to INT4), which can lead to substantial compression. However, hardware-dependent support for quantization techniques varies across platforms, potentially limiting model portability. Knowledge distillation transfers knowledge from a large teacher model to a smaller student model by aligning their behaviors. While effective, it often requires costly and time-consuming redesign of the student model (Gou et al., 2021; Eldan & Li, 2023). In contrast, structural pruning removes redundant model components (e.g., neurons, channels, or attention heads) (Michel et al., 2019; Frantar & Alistarh, 2022; Zhu et al., 2024), thereby reducing both memory usage and floating-point operations (FLOPs). Its key advantage is enabling hardware-agnostic inference acceleration, making it a promising direction in model compression research.

Traditional structural pruning methods (Zafrir et al., 2021; Liu et al., 2021) typically decouple pruning and fine-tuning, following either a pruning-then-finetuning or finetuning-then-pruning paradigm. This disjointed process disrupts the synergy between pruning and task adaptation, leading to notable

and often irreversible performance degradation. Such degradation arises because pruning disrupts the knowledge embedded in pre-trained weights, while subsequent fine-tuning fails to fully recover the lost capacity. To address this issue, Liu et al. (2025) recently proposed a novel paradigm called Pruning-Aware Tuning (PAT). By integrating pruning and fine-tuning into a unified framework, PAT allows the model to simultaneously adapt to downstream tasks and identify redundant parameters. This collaborative learning strategy not only alleviates performance degradation but, in some cases, even allows pruned models to surpass the performance of their unpruned counterparts.

However, most existing pruning methods, including PAT, apply uniform pruning strategies across all layers or modules of the model. This implicitly assumes that redundancy is evenly distributed throughout the model, overlooking the layer-wise functional diversity and risking the removal of essential model components. Extensive research (Tenney et al., 2019; Belinkov & Glass, 2019; Geva et al., 2020) has demonstrated that deep neural networks, including LLMs, learn hierarchical feature representations. Lower layers typically capture general linguistic patterns such as syntactic structures, while higher layers tend to encode more abstract, task-specific semantic and logical information. Although some studies (Sun et al., 2024; Ling et al., 2024; Ali et al., 2025) have attempted to differentiate pruning strategies across layers, they often rely on pre-defined heuristic metrics to determine pruning targets. These approaches cannot support end-to-end optimization for learning task-driven, adaptive sparsity directly from the model's own gradients and data flow.

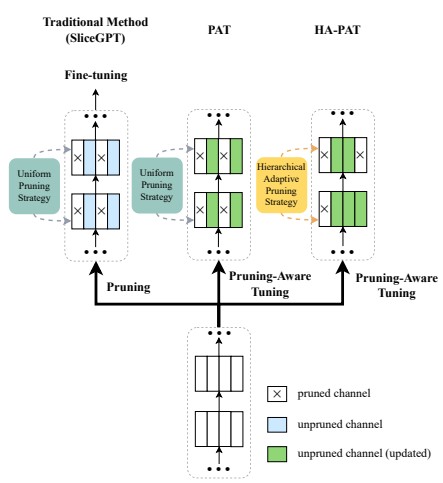

Figure 1: Comparisons between our method (HA-PAT), traditional structured pruning (e.g., SliceGPT), and PAT. (1) Traditional methods (e.g., SliceGPT) apply uniform pruning strategy across all modules and require separate fine-tuning; (2) PAT integrates pruning into fine-tuning but uses uniform strategiey for all modules; (3) HA-PAT performs pruning during fine-tuning and adapts layer-specific pruning strategiey.

To tackle these challenges, we propose Hierarchically-Adaptive Pruning-Aware Tuning (HA-PAT), a flexible and structure-aware paradigm that integrates hierarchical adaptivity into the pruning-aware fine-tuning framework. A conceptual comparison of HA-PAT against related methods is presented in 1. Specifically, we introduce three key improvements over the PAT paradigm: (1) Layer-wise Independent Masks (LIM), which replace the uniform mask strategy, enabling each module to independently learn its optimal pruning structure; (2) Hierarchical Pruning Ratio Scheduling (HPRS), which applies progressively increasing pruning ratios across model depths, thereby better capturing the layer-wise distribution of redundancy in LLMs; (3) Adaptive Compensation Operator (ACO), which decouples the pre-trained weights into magnitude and direction, enabling the reconstruction of pruned weights based on a richer representational space.

Extensive experiments on multiple LLMs demonstrate that our proposed HA-PAT framework achieves significant performance improvements over baseline methods. Under a 25% pruning ratio on the LLaMA2-13B model, HA-PAT outperforms the PAT approach by 4.01%, along with a 30% inference speedup. The main contributions of this work can be summarized as follows:

- We propose the HA-PAT framework to solve the performance bottleneck caused by existing uniform pruning strategies that ignore layer-wise functional diversity.

- We introduce Layer-wise Independent Masks (LIM), enabling each layer to autonomously learn its optimal sparse structure , thus facilitating more fine-grained and flexible pruning.

- We design Hierarchical Pruning Ratio Scheduling (HPRS) to preserve foundational language representations in shallow layers while more aggressively compressing redundant, task-specific features in deeper layers.

## 2 RELATED WORK

### 2.1 PRUNING

Network pruning is a fundamental and widely used technique in model compression, aimed at reducing computational and storage costs by removing redundant parameters. According to the level of granularity, pruning methods are generally classified into unstructured pruning and structured pruning. Unstructured pruning removes individual weights based on importance metrics. For instance, Wanda (Sun et al., 2024) and SparseGPT(Frantar & Alistarh, 2023) identify and remove the least important weights using weight-activation products and Hessian information, respectively. Although these methods can achieve high sparsity, the resulting irregular sparse patterns typically require specialized hardware or custom computational libraries for efficient inference. In contrast, structured pruning has garnered increasing attention for its hardware-friendliness, as it removes entire parameter groups (e.g., rows or columns of weight matrices). For example, LLM-Pruner (Ma et al., 2023) leverages gradient-based dependency analysis for task-aware structured pruning, while Sheared LLAMA (Xia et al., 2023) performs end-to-end shape-oriented pruning with dynamic batch loading to enhance efficiency. However, existing structured pruning methods predominantly follow a pruning-then-finetuning paradigm. This disjointed process disrupts the knowledge embedded in pre-trained weights, leading to significant and often irreversible performance degradation.

### 2.2 PARAMETER-EFFICIENT FINE-TUNING

Parameter-Efficient Fine-Tuning (PEFT) aims to achieve performance comparable to full fine-tuning while updating only a small subset of model parameters or introducing a limited number of trainable modules, thereby significantly reducing computational and storage costs. Among various approaches, Low-Rank Adaptation (LoRA) (Hu et al., 2022) is one of the most representative. LoRA freezes all pre-trained weights and injects two trainable low-rank matrices into specific linear layers. During fine-tuning, only these low-rank matrices are updated, substantially decreasing the number of trainable parameters. Building on this foundation, recent methods such as DoRA (Liu et al., 2024a) further enhance performance by decomposing pre-trained weights into magnitude and direction components. However, it is important to note that most PEFT methods are primarily designed to reduce training costs. During inference, they still require to load the full model, thus failing to reduce the model's storage requirements and inference latency, which are advantages that structured pruning can effectively offer.

## 3 METHODOLOGY

Our work builds upon and significantly extends the Pruning-Aware Tuning (PAT) paradigm (Liu et al., 2025). The following subsections will first introduce the PAT framework as a foundation for our work, and then detail our hierarchically-adaptive enhancements designed to address its inherent limitations.

### 3.1 PRELIMINARY: PRUNING-AWARE TUNING (PAT)

The core idea of PAT is to synchronously and adaptively remove redundant structures in LLMs during fine-tuning. This is achieved by inserting pluggable Hybrid Sparsification Modules (HSMs) between the Attention and FFN components. Each HSM operates on the output of its preceding linear layer and consists of a compensation matrix $\mathbf{D}$ and a trainable pruning mask $\mathbf{m}$. The computation can be formulated as:

$$
\begin{aligned}
\mathbf{Z} &= (\mathbf{m} \odot \mathbf{D}) \cdot \mathbf{W}\mathbf{X} \\
&= (\mathbf{m} \odot \mathbf{D}\mathbf{W}) \cdot \mathbf{X}
\end{aligned}
\tag{1}
$$

where $\mathbf{X} \in \mathbf{R}^{d_i \times n}$ is the input, $\mathbf{W} \in \mathbf{R}^{d_o \times d_i}$ denotes the weights of the upstream linear layer, $d_o$ and $d_i$ are the output and input dimensions of the layer, and $\odot$ indicates element-wise multiplication. $\mathbf{D}$ is a trainable compensation matrix, and $\mathbf{m} \in \mathbf{R}^{d_o}$ is a pruning mask whose values converge to 0 or 1 after training, determining which output channels are retained. To reduce training overhead, $\mathbf{D}$ is constructed as a parameter-efficient Hybrid Identity Operator (HIO), defined as:

$$
\mathbf{D} = \mathbf{L}_1 \cdot \mathbf{L}_0 + \mathbf{I}
\tag{2}
$$

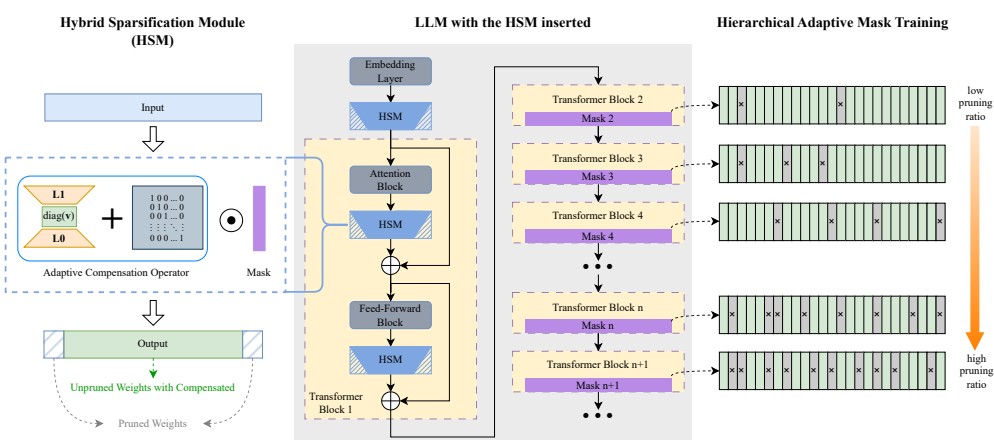

Figure 2: Framework of Hierarchically-Adaptive Pruning-Aware Tuning (HA-PAT). Similar to the PAT paradigm, HA-PAT inserts Hybrid Sparsification Modules (HSMs) after the Attention and Feed-Forward Network. Each HSM consists of an Adaptive Compensation Operator (ACO) and an independent trainable pruning mask. Leveraging Hierarchical Pruning Ratio Scheduling (HPRS), HA-PAT performs channel-wise pruning in a layer-adaptive manner, aligned with the functional heterogeneity across different layers of LLMs. During fine-tuning, both the masks and compensation matrices are jointly optimized. At inference stage, the learned compensation matrices are merged into the upstream linear layers, and redundant channels are removed based on the mask values.

where $\mathbf{L_0} \in \mathbf{R}^{r \times d_o}$, $\mathbf{L_1} \in \mathbf{R}^{d_o \times r}$, $r$ is the rank of the low-rank component, and $\mathbf{I} \in \mathbf{R}^{d_o \times d_o}$ is the identity matrix. During training, $\mathbf{I}$ remains fixed, allowing gradients to flow only through $\mathbf{L_0}$ and $\mathbf{L_1}$. During fine-tuning, both the mask and compensation matrices are jointly optimized. At inference stage, the learned compensation matrices are merged into the upstream linear layers, and redundant channels are removed based on the mask values.

However, a key characteristic of PAT is the Unified Sparsification Mask (USM), where all HSMs across the model share a single, globally trained mask $\mathbf{m}$. While this design ensures structural consistency, it imposes a fundamental limitation: it enforces a uniform sparsity pattern across all layers. This strategy implicitly assumes that model redundancy is evenly distributed, thereby overlooking the functional heterogeneity present in different layers of LLMs.

Motivated by this, we propose Hierarchically-Adaptive Pruning-Aware Tuning (HA-PAT). HA-PAT introduces hierarchical adaptivity through Layer-wise Independent Masks (LIM) and Hierarchical Pruning Ratio Scheduling (HPRS), enabling layer-specific sparsity that better aligns with the representational roles of each layer. The overall framework of HA-PAT is illustrated in Fig. 2.

## 3.2 Layer-wise Independent Masks (LIM)

A straightforward strategy for introducing hierarchical adaptivity is to relax the global sharing constraint imposed by the USM. We propose Layer-wise Independent Masks (LIM), which applies a more flexible mechanism tailored to the hierarchical heterogeneity of LLMs. Specifically, each HSM is assigned an independent, trainable mask $\mathbf{m}_l$, where $l$ denotes the layer index. This empowers each layer to autonomously learn its optimal sparse structure based on its specific functional role and the gradients it receives from the task loss. Accordingly, Eq. (1) is reformulated as:

$$\mathbf{Z}_l = (\mathbf{m}_l \odot \mathbf{D}_l \mathbf{W}_l) \cdot \mathbf{X}_l \tag{3}$$

where $\mathbf{Z}_l, \mathbf{m}_l, \mathbf{D}_l, \mathbf{W}_l, \mathbf{X}_l$ are the corresponding variables for layer $l$. LIM provides the necessary degrees of freedom for each layer to adapt its structure, enabling a more granular, layer-aware pruning strategy. This fine-grained control, attained with only minimal training overhead, constitutes a necessary prerequisite for enabling a structured and hierarchical pruning paradigm, as elaborated in the subsequent section.

### 3.3 HIERARCHICAL PRUNING RATIO SCHEDULING (HPRS)

Although LIM offers the necessary flexibility, it implicitly assumes all layers are equally important and lack prior guidance for optimization. This can lead to inefficient exploration of the optimal sparse structure, as the model must learn the relative importance of each layer from scratch. To address this, we introduce Hierarchical Pruning Ratio Scheduling (HPRS). Instead of enforcing a uniform pruning ratio, it applies a structural pruning schedule where the target pruning ratio $p_l$ for the $l$-th layer increases progressively with depth.

The Information Bottleneck (IB) principle(Tishby et al., 2000) posits that each layer of a deep neural network learns an optimal representation $Z$, that maximizes compression of its input $X$ while preserving as much information as possible about the final task $Y$. This trade-off is formally captured by the following objective:

$$\min_{p(z|x)} \mathcal{L}(Z) = I(X;Z) - \beta I(Z;Y) \tag{4}$$

where $I(\cdot;\cdot)$ denotes mutual information, and $\beta$ is a Lagrange multiplier that balances the trade-off between compression and prediction. Furthermore, an $L$-layer language model can be seen as an information-processing Markov chain:

$$X \to Z_1 \to \cdots \to Z_l \to \cdots \to Z_L \to \hat{Y} \tag{5}$$

where $X$ is the input text, $Z_l$ is the hidden representation of the $l$-th layer, and $\hat{Y}$ is the final predicted output. According to the Data Processing Inequality(Cover, 1999),

$$I(X;Y) \geq I(Z_1;Y) \geq I(Z_2;Y) \geq \cdots \geq I(Z_L;Y) \tag{6}$$

which means that the mutual information with respect to the target task $Y$ cannot increase as information is passed through successive layers. The role of each layer is to refine and transform the information, making its representation more suitable for processing by subsequent layers and ultimately for making predictions.

The shallow layers of the model extract fundamental linguistic features (e.g., syntax) essential for all downstream tasks, thus their representations $Z_l$ must maximize information about the input $I(X;Z_l)$. Structural pruning directly constrains this information flow by setting a physical upper bound via the channel capacity $C_l$, where $I(X;Z_l) \leq C_l$. Aggressive pruning in shallow layers would sharply decrease $C_l$, creating a restrictive bottleneck that causes an irreversible loss of this foundational information, crippling the reasoning capabilities of subsequent layers. Therefore, a low pruning ratio is crucial to preserve the model's core linguistic capabilities. In contrast, the deep layers are more specialized, focusing on refining abstract features relevant to the specific downstream task $Y$ and aiming to maximize $I(Z_l;Y)$. To achieve better generalization, the IB principle suggests these layers should compress their input $Z_{l-1}$ by retaining only task-relevant information. Moreover, Sun et al. (2025b;a); Gromov et al. (2024) have revealed that deep layers in LLMs often suffer from diminished effectiveness. Applying a higher pruning ratio in deep layers thus serves as a reasonable and effective mechanism for this task-oriented compression, aligning with the model's optimization goals.

Therefore, the HPRS strategy, which employs conservative pruning in shallow layers and aggressive pruning in deep layers, constitutes a direct and practical application of the Information Bottleneck principle in the context of structured pruning for large language models. We instantiate this strategy with a simple yet effective linear scheduling function. Specifically, the target pruning ratio $p_l$ for the $l$-th layer is defined as:

$$p_l = p_{\min} + (p_{\max} - p_{\min}) \cdot \frac{l-1}{L-1} \tag{7}$$

where $L$ denotes the total number of transformer modules, $l \in \{1, 2, ..., L\}$ is the layer index, and $p_{\min}, p_{\max}$ are target ratios for the first and last layers, respectively. The sensitivity analysis of the HPRS pruning ratio range are detailed in Sec. A.4. To validate that the effectiveness of HPRS stems from the hierarchical principle itself rather than this specific linear formulation, we also explored several non-linear scheduling functions (e.g., cosine and sigmoid). These comprehensive robustness analyses and experiments, detailed in Sec. A.2 and Sec. A.3. To enforce this schedule during training, we define the pruning ratio regularization term $\mathcal{L}_{ratio}$ in a layer-wise manner:

$$\mathcal{L}_{ratio} = \sum_{l=0}^{L-1} \left\| N_{target,l} - \sum_{i=1}^{d_h} \mathbf{1}(m_{i,l} > 0) \right\|_2 \tag{8}$$

where $m_{i,l}$ is the $i$-th element of $\mathbf{m}_l$, $d_h$ is the hidden dimension, $\mathbf{1}(\cdot)$ is the indicator function, and $N_{target,l}$ denotes the target number of retained channels for layer $l$. This regularization term encourages each layer to converge to its designated sparsity level. By combining the flexibility of LIM with the structured guidance of HPRS, HA-PAT can more precisely identify and target redundant structures within the model.

## 3.4 ADAPTIVE COMPENSATION OPERATOR (ACO)

In addition to enabling HA-PAT to adaptively identify layer-wise redundancy, we also aim to mitigate performance degradation caused by pruning. Inspired by DoRA (Liu et al., 2024a), we posit that an effective compensation process should decompose the complex task of knowledge transfer into two orthogonal subproblems: redirecting the flow of information (direction) and modulating the signal strength (magnitude). Decoupling these factors leads to more stable optimization and more efficient knowledge transfer. Therefore, we propose the Adaptive Compensation Operator (ACO), which achieves functional decoupling by introducing a trainable scaling vector $\mathbf{v} \in \mathbf{R}^r$ (where $r$ is the rank of low-rank matrices) between the two low-rank matrices $\mathbf{L_0}$ and $\mathbf{L_1}$ in HIO. The modified compensation matrix $\mathbf{D}'$ is:

$$\mathbf{D}' = \mathbf{L_1} \cdot \mathrm{diag}(\mathbf{v}) \cdot \mathbf{L_0} + \mathbf{I} \tag{9}$$

where $\mathrm{diag}(\mathbf{v})$ transforms $\mathbf{v}$ into a diagonal matrix. Under this parameterization paradigm, (1) the matrices $\mathbf{L_0}$ and $\mathbf{L_1}$ are encouraged to become orthogonal under the constraint of identity loss $\mathcal{L}_{identity}$ (See Eq. (11)). Their primary role is to learn an orthogonal transformation basis, which represents the direction of knowledge transfer. (2) the vector $\mathbf{v}$ explicitly learns the importance of each basis vector (i.e., each rank component), representing the magnitude of the knowledge transfer. This decoupled direction-magnitude parameterization offers greater expressiveness and better optimization characteristics than the original HIO. It enables the model to more robustly and precisely reorganize and transfer knowledge from pruned channels to the retained ones, ultimately enhancing the overall performance of the pruned model.

## 3.5 OVERALL OPTIMIZATION OBJECTIVE

The training of HA-PAT is governed by a multi-objective loss function that jointly optimizes for task performance, target sparsity, and training stability. The overall optimization objective $\mathcal{L}$ is defined as:

$$\mathcal{L} = \mathcal{L}_{instruct} + \mathcal{L}_{ratio} + \mathcal{L}_{identity} \tag{10}$$

Here, $\mathcal{L}_{instruct}$ is the standard instruction fine-tuning loss, typically cross-entropy. It serves as the primary driver for task-specific learning, ensuring that both pruning decisions and information recovery are ultimately aimed at maximizing task-relevant information. Minimizing cross-entropy is equivalent to minimizing the conditional entropy $H(Y|Z)$, which in turn is equivalent to maximizing the mutual information $I(Z;Y)$. Consequently, the process of minimizing $L_{instruct}$ via gradient descent is, in essence, implicitly maximizing the mutual information $I(Z;Y)$ between the model's representations and the task labels. Since the learning of the pruning masks $M_l$ is driven by these same gradients, the model adaptively learns to retain neural channels that are most critical for maximizing $I(Z;Y)$. The other two terms are regularizers: $\mathcal{L}_{ratio}$ imposes cumulative sparsity constraints across the network layers (see Eq. (8)); $\mathcal{L}_{identity}$ imposes an orthogonality constraint on the low-rank factors $\mathbf{L_0}$ and $\mathbf{L_1}$ within the ACO, ensuring they specialize in learning the direction of knowledge transfer, while stabilizing the gradient flow and the overall learning process. Specifically, the identity regularization term is defined as:

$$\mathcal{L}_{identity} = \sum_{l=0}^{L-1} (\|\mathbf{L_{0},}_l \cdot \mathbf{L_{0,}}_l^{\mathbf{T}} - \mathbf{I}\|_2 + \|\mathbf{L_{1,}}_l^{\mathbf{T}} \cdot \mathbf{L_{1,}}_l - \mathbf{I}\|_2) \tag{11}$$

where $\mathbf{L_{0},}_l$ and $\mathbf{L_{1},}_l$ are low-rank factors forming $\mathbf{D}_l$.

## 4 EXPERIMENTS

### 4.1 EXPERIMENTAL SETUP

**Models.**   We conduct experiments on four widely adopted large language models: Gemma-2B and 7B (Team et al., 2024), Llama2-7B and 13B (Touvron et al., 2023a). For each model, we evaluate our method under three pruning ratios: 20%, 25%, and 30%.

**Baselines.**   In addition to the PAT, we compare against two representative structured pruning approaches: LLM-Pruner (Ma et al., 2023) and SliceGPT (Ashkboos et al., 2024), both of which follow a pruning-then-finetuning pipeline. We also include standard LoRA fine-tuning without pruning as a performance reference.

**Datasets.**   We use the LaMini-instruction dataset (Wu et al., 2023) for supervised fine-tuning. To reduce training costs, we randomly sample 10% of the datasets, resulting in 256K training pairs. For evaluation, we follow the PAT paper and test on 14 downstream tasks using zero-shot accuracy with the First-Capital-Word method (Contributors, 2023).

**Implementation Details.**   All experiments are conducted on NVIDIA A100 GPUs for 3 epochs. Following the PAT configuration, we use a learning rate of $5 \times 10^{-5}$ and apply cosine annealing to schedule the mask convergence within the first third of the training steps. The batch size is set to 128, and the sequence length is 256 tokens.

### 4.2 EXPERIMENTAL RESULTS AND ANALYSIS

**Performance Comparison.**   Tab. 1 shows the average zero-shot accuracy of HA-PAT, compared with our reproduced PAT and other structured pruning methods across 14 downstream tasks. The results reveal the following key findings: (1) our proposed HA-PAT consistently outperforms the PAT baseline in terms of average accuracy, strongly supporting our central claim that incorporating a hierarchical adaptive mechanism leads to improving model performance; (2) the model pruned by HA-PAT exhibits minimal performance loss and, surprisingly, even outperforms the unpruned LoRA baseline in certain cases. For instance, under a 20% pruning ratio, the HA-PAT-pruned LLaMA2 model achieves 2% to 4% higher performance than the unpruned LoRA baseline. See the Sec. A.8 for detailed results on each dataset.

Table 1: Zero-shot evaluation results of different pruning methods on four LLMs with 20%, 25%, and 30% pruning ratios. The structured pruning methods LLM-Pruner and SliceGPT both adopt the pruning-then-finetuning paradigm. Accuracy is the average across 14 datasets.

| Ratio | Method | Gemma 2B | Gemma 7B | Llama2 7B | Llama2 13B |
|---|---|---|---|---|---|
| **0%** | LoRA-64 | 53.82 | 71.59 | 58.76 | 66.74 |
| **20%** | LLM-Pruner | 48.87 | 65.45 | 58.53 | 65.28 |
| | SliceGPT | 48.21 | 66.60 | 57.81 | 65.86 |
| | PAT | 48.56 | 65.70 | 58.84 | 65.64 |
| | **Ours** | **53.84** | **68.32** | **63.06** | **68.96** |
| **25%** | LLM-Pruner | 42.32 | 60.50 | 52.50 | 58.63 |
| | SliceGPT | 45.23 | 62.22 | 52.98 | 60.69 |
| | PAT | 47.79 | 61.43 | 56.52 | 63.36 |
| | **Ours** | **49.65** | **63.75** | **59.88** | **67.37** |
| **30%** | LLM-Pruner | 39.71 | 50.28 | 50.60 | 51.28 |
| | SliceGPT | 40.07 | 53.14 | 50.91 | 56.12 |
| | PAT | 45.92 | 59.14 | 56.32 | 58.63 |
| | **Ours** | **49.04** | **60.43** | **58.74** | **65.69** |

Table 2: Ablation study on Layer-wise Independent Masks (LIM), Hierarchical Pruning Ratio Scheduling (HPRS) and Adaptive Compensation Operator (ACO) for Llama2 models.

| Ratio | Metric | Llama2 7B | | | | | Llama2 13B | | | | |
|---|---|---|---|---|---|---|---|---|---|---|---|
| | | PAT | Ours | | | | PAT | Ours | | | |
| 20% | LIM | ✗ | ✓ | ✓ | ✗ | ✓ | ✗ | ✓ | ✓ | ✗ | ✓ |
| | HPRS | ✗ | ✗ | ✓ | ✗ | ✓ | ✗ | ✗ | ✓ | ✗ | ✓ |
| | ACO | ✗ | ✗ | ✗ | ✓ | ✓ | ✗ | ✗ | ✗ | ✓ | ✓ |
| | Acc. | 58.84 | 61.16 | 62.83 | 60.03 | **63.06** | 65.64 | 68.24 | 68.85 | 66.53 | **68.96** |
| 25% | LIM | ✗ | ✓ | ✓ | ✗ | ✓ | ✗ | ✓ | ✓ | ✗ | ✓ |
| | HPRS | ✗ | ✗ | ✓ | ✗ | ✓ | ✗ | ✗ | ✓ | ✗ | ✓ |
| | ACO | ✗ | ✗ | ✗ | ✓ | ✓ | ✗ | ✗ | ✗ | ✓ | ✓ |
| | Acc. | 56.52 | 58.30 | 58.38 | 58.13 | **59.88** | 63.36 | 66.91 | 67.09 | 64.01 | **67.37** |

Table 3: Ablation study on Layer-wise Independent Masks (LIM), Hierarchical Pruning Ratio Scheduling (HPRS) and Adaptive Compensation Operator (ACO) for Gemma models.

| Ratio | Metric | Gemma 2B | | | | | Gemma 7B | | | | |
|---|---|---|---|---|---|---|---|---|---|---|---|
| | | PAT | Ours | | | | PAT | Ours | | | |
| 20% | LIM | ✗ | ✓ | ✓ | ✗ | ✓ | ✗ | ✓ | ✓ | ✗ | ✓ |
| | HPRS | ✗ | ✗ | ✓ | ✗ | ✓ | ✗ | ✗ | ✓ | ✗ | ✓ |
| | ACO | ✗ | ✗ | ✗ | ✓ | ✓ | ✗ | ✗ | ✗ | ✓ | ✓ |
| | Acc. | 48.56 | 49.88 | 52.46 | 51.85 | **53.84** | 65.70 | 66.05 | 67.54 | 68.09 | **68.32** |
| 25% | LIM | ✗ | ✓ | ✓ | ✗ | ✓ | ✗ | ✓ | ✓ | ✗ | ✓ |
| | HPRS | ✗ | ✗ | ✓ | ✗ | ✓ | ✗ | ✗ | ✓ | ✗ | ✓ |
| | ACO | ✗ | ✗ | ✗ | ✓ | ✓ | ✗ | ✗ | ✗ | ✓ | ✓ |
| | Acc. | 47.79 | 48.13 | 49.05 | 48.32 | **49.65** | 61.43 | 61.84 | 62.45 | 63.32 | **63.75** |

**Ablation Analysis.** To validate the effectiveness of each core component in our framework, we conduct a series of ablation experiments. From Tab. 2 and Tab. 3, we get the following key observations: (1) layer-wise adaptive masking serves as the foundation of HA-PAT's performance improvements. By empowering each layer to learn its optimal sparse structure, it consistently outperforms the baseline that uses a globally shared mask; (2) building upon LIM's strong adaptive capacity, HPRS provides stable and consistent performance gains. As a fine-grained control mechanism, HPRS helps guide the model toward more favorable sparse solutions, further optimizing performance beyond what LIM alone can achieve; (3) HA-PAT variants with only ACO also consistently outperform the PAT baseline, as its direction-magnitude decoupling enhances functional transferability. Ultimately, the combination of HPRS and ACO works synergistically, tackling both the identification of redundancy and the facilitation of functional transfer to achieve precise and efficient model pruning.

**Downstream Task Capability Analysis.** Although HA-PAT consistently demonstrates superior average performance, its improvements are not uniformly distributed across all downstream tasks. To investigate this phenomenon in depth, we analyze the performance differences between HA-PAT and the PAT baseline across various tasks, as shown in Fig. 3. A clear pattern emerges: HA-PAT shows the most significant advantages on tasks that requiring deep language understanding, such as WSC (which requires precise coreference resolution), BOOLQ (which involves textual information extraction), and MultiRC (which demands multi-sentence reasoning). In contrast, for tasks that heavily rely on commonsense reasoning, the advantages of HA-PAT become less prominent and, in some cases, it even underperforms the PAT baseline, as observed on WinoGrande (WG) and PIQA.

This phenomenon underscores the intrinsic alignment between pruning strategies and the characteristics of specific tasks. HPRS tends to preserve general linguistic representations in shallow layers, while guiding deeper layers to specialize in more abstract, task-relevant semantic functions. Consequently, for tasks such as WSC, where precise disambiguation and contextual understanding are

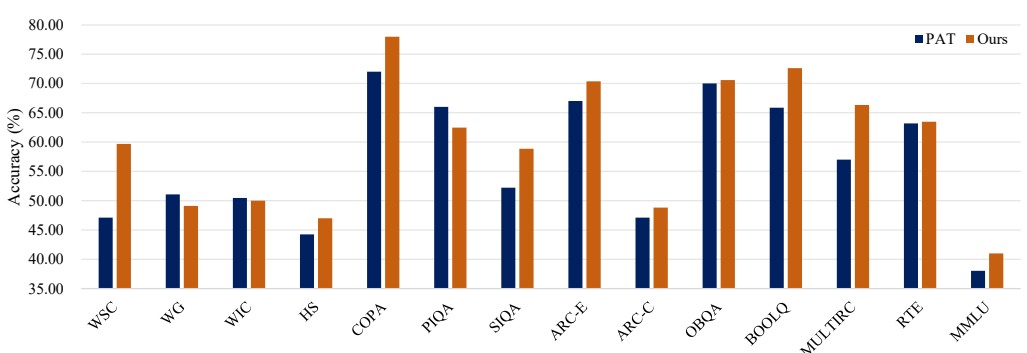

Figure 3: Performance comparison of the Llama2-7B model across 14 tasks under a 25% pruning ratio.

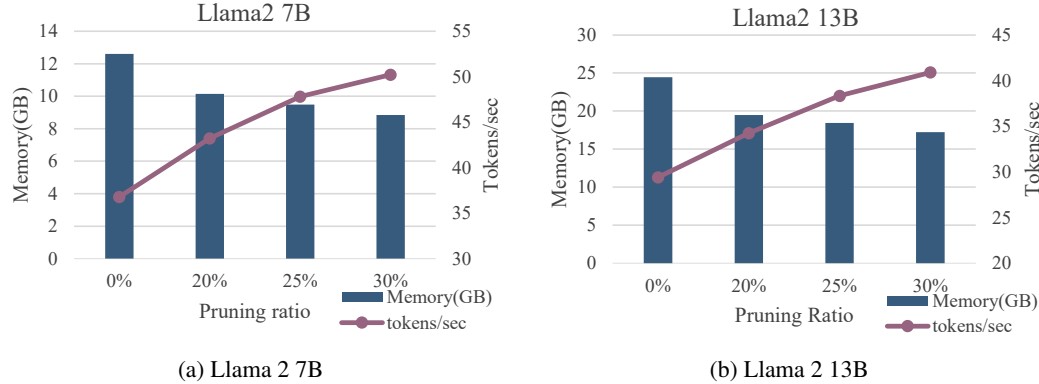

(a) Llama 2 7B                      (b) Llama 2 13B

Figure 4: The VRAM usage and speedup of Llama2 models under various pruning ratios.

essential, HA-PAT demonstrates superior effectiveness. Conversely, for tasks that rely more on associative knowledge distributed throughout the network, the uniform pruning strategy of PAT may better preserve globally relevant information. These task-specific variations suggest that HA-PAT is more effective at preserving the model's language understanding capabilities. This finding further supports the intrinsic alignment between the hierarchical structure of LLMs and their functional specialization.

**Memory and Latency.** Beyond improvements in model accuracy, HA-PAT also inherits the efficiency benefits of structured pruning. As shown in Fig. 4, applying a 25% pruning ratio to the LLaMA2-13B model reduces GPU memory usage by approximately 6.01 GB compared to the original dense model and improves average inference speed by 30%. These results demonstrate that HA-PAT is not only theoretically effective but also offers a practical solution for deploying large language models on resource-constrained devices.

## 5 CONCLUSION

We propose the Hierarchically-Adaptive Pruning-Aware Tuning (HA-PAT). HA-PAT integrates pruning and fine-tuning into a unified process, enabling adaptive pruning that aligns with the layerwise functional diversity by introducing Layer-wise Independent Masks (LIM) and Hierarchical Pruning Ratio Scheduling (HPRS). This allows for precise identification and removal of redundant structures during fine-tuning. Moreover, the Adaptive Compensation Operator (ACO) further enhances the model's ability to recover performance after pruning. Experimental results show that HA-PAT significantly improves inference efficiency while maintaining the performance of the unpruned baselines, particularly on tasks requiring deep language understanding. These findings underscore

the importance of task-aware adaptive pruning strategies and offer an efficient and high-performing solution for the practical deployment of LLMs.

**Limitations and future work:** Although HA-PAT demonstrates strong theoretical grounding and empirical gains, several limitations suggest promising directions for future research. First, the HPRS schedule, though theoretically justified via the Information Bottleneck and empirically robust to functional variations, is still predefined. Future work could investigate learnable scheduling strategies for fully adaptive end-to-end pruning. Second, our analysis indicates that a single hierarchical pruning strategy may not be universally optimal for all types of downstream tasks. Future research could explore task-specific adaptive pruning mechanisms that dynamically adjust strategies based on task characteristics. Furthermore, applying hierarchical adaptation to other structural components, or extending our framework to accommodate the intertwined feature representations observed in Loop Transformers, presents a promising avenue for further investigation.

## REPRODUCIBILITY STATEMENT

We are committed to ensuring the full reproducibility of our research. The overall pipeline of our proposed HA-PAT method is detailed in Algorithm 1. For complete transparency and to facilitate reproduction, our entire source code and experimental scripts are included in the supplementary materials. Comprehensive details of our experimental setup, including the models used, data processing procedures, and all hyperparameters (e.g., learning rate, batch size, and the $p_{\min}$ and $p_{\max}$ settings for HPRS), are thoroughly documented in Sec. 4.1 and Sec. A.6. All datasets used for fine-tuning (LaMini-instruction) and evaluation (14 downstream benchmarks) are publicly available resources, with corresponding citations provided in Sec. 4.1 and Sec. A.6. Furthermore, our experimental framework is built upon publicly available and widely adopted open-source components to maximize transparency and ease of verification. The base models we use (Llama2, Gemma) are sourced from the Hugging Face Hub, ensuring a standardized starting point for all experiments.

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

# A APPENDIX

## A.1 USE OF LARGE LANGUAGE MODELS (LLMS)

During the preparation of this paper, we used large language models (LLMs) as a writing assistant. The LLMs were used to enhance the clarity, grammar, and fluency of the text, ensuring it adheres to a formal academic writing style.

## A.2 HPRS SCHEDULING STRATEGIES

In the Sec. 3.3, we instantiate the Hierarchical Pruning Ratio Scheduling (HPRS) with a simple linear function for clarity. To demonstrate that the effectiveness of HPRS stems from its hierarchical pruning principle rather than a specific functional form, we introduce four alternative scheduling strategies. All strategies compute a target pruning ratio $p_l$ for each layer $l$ (where $l \in \{1, 2, \ldots, L\}$), which progressively increases from a predefined minimum value $p_{\min}$ to a maximum value $p_{\max}$.

**1. Linear Scheduling** This is the default strategy used in the main experiments, providing a uniform increase in the pruning ratio with layer depth:

$$p_l = p_{\min} + (p_{\max} - p_{\min}) \cdot \frac{l - 1}{L - 1} \tag{12}$$

**2. Half-Cosine Scheduling-1** This schedule uses a quarter of a cosine period with a fast-start and slow-finish pattern, where the growth of the pruning ratio decelerates with layer depth:

$$p_l = p_{\max} + (p_{\min} - p_{\max}) \cdot (1 + \cos \frac{\pi}{2}(1 + \frac{l-1}{L-1})) \tag{13}$$

**3. Half-Cosine Scheduling-2** This schedule uses a quarter of a cosine period with a slow-start and fast-finish pattern, where the growth of the pruning ratio accelerates with layer depth:

$$p_l = p_{\max} + (p_{\min} - p_{\max}) \cdot \cos \frac{\pi(l-1)}{2(L-1)} \tag{14}$$

**4. Cosine Scheduling** This schedule follows half of a full cosine period, yielding an S-shaped pruning ratio growth curve that is slow at the beginning and end, and fast in the middle:

$$p_l = p_{\max} + \frac{p_{\min} - p_{\max}}{2} \cdot (1 + \cos \frac{\pi(l-1)}{L-1}) \tag{15}$$

**5. Sigmoid Scheduling** This is a classic S-shaped function where the pruning ratio increases rapidly around the middle layers and more slowly at the extremes:

$$p_l = p_{\max} + (p_{\min} - p_{\max}) \cdot \frac{1}{1 + e^{k(\frac{l-1}{L-1} - \frac{1}{2})}} \tag{16}$$

In all the formulations above, $L$ denotes the total number of transformer modules in the model, $l$ is the current layer index, where $l \in \{1, 2, \ldots, L\}$, $p_{\min}$ and $p_{\max}$ are the target pruning ratios for the first and last layers respectively, and $k$ is the slope parameter of the sigmoid (which is set to 12 in our experiments). The performance comparison of these scheduling strategies is presented in Sec. A.3, confirming the generality of the HPRS principle.

### A.3 ROBUSTNESS ANALYSIS OF HPRS SCHEDULING STRATEGIES

To verify that the performance gains of HA-PAT primarily derive from the hierarchical principle, rather than a specific scheduling function, we evaluate the five strategies described in Sec. A.2. The experiments are conducted with Llama2 7B under two pruning ratio settings: 25%±5% and 30%±5%. The results are shown in Tab. 4.

Table 4: Comparison of average zero-shot accuracy on 14 downstream tasks for different HPRS scheduling strategies.

| Ratio | PAT | Ours | | | | |
|---|---|---|---|---|---|---|
| | | Linear | Half-Cosine 1 | Half-Cosine 2 | Cosine | Sigmoid |
| **25%±5%** | 56.52 | 59.88 | 60.82 | 59.95 | 60.53 | 58.72 |
| **30%±5%** | 56.32 | 58.74 | 56.85 | 58.78 | 57.24 | 58.37 |

The results lead to two key findings. First, all scheduling strategies significantly outperform the PAT baseline which uses uniform pruning, confirming the universal effectiveness of the hierarchical approach. Second, despite their different functional forms (linear, fast-start, slow-start, S-shaped), the performances of these five strategies are closely aligned, suggesting that the success of HA-PAT stems from the hierarchical principle itself rather than careful tuning of the scheduling curve.

### A.4 SENSITIVITY ANALYSIS OF THE HPRS PRUNING RATIO RANGE

Another critical hyperparameter of HPRS is the pruning ratio range $\Delta p$, which determines the gap between $p_{\min}$ and $p_{\max}$, i.e., the steepness of the hierarchy. To investigate the sensitivity of model performance to this hyperparameter, we centered the pruning ratio $p$ at 25% and 30% and varied $\Delta p$ within the set $\{5\%, 10\%, 15\%\}$. The results are presented in Tab. 5.

Table 5: Comparison of average zero-shot accuracy of HA-PAT on Llama2 7B under different pruning ranges $\Delta p$. The HPRS strategy is Linear Scheduling function.

| Center Ratio ($p$) | Method | Range ($\Delta p$) | $p_{\min}$ | $p_{\max}$ | Accuracy |
|---|---|---|---|---|---|
| | **PAT** | N/A | N/A | N/A | 56.52 |
| **25%** | | 5% | 20% | 30% | 59.88 |
| | **Ours** | 10% | 15% | 35% | 60.68 |
| | | 15% | 10% | 40% | 61.30 |
| | **PAT** | N/A | N/A | N/A | 56.32 |
| **30%** | | 5% | 25% | 35% | 58.74 |
| | **Ours** | 10% | 20% | 40% | 59.12 |
| | | 15% | 15% | 45% | 60.72 |

Two observations emerge. First, our method consistently outperforms the uniform-pruning PAT baseline across all tested $\Delta p$ settings, underscoring the superiority of hierarchical pruning. Second, we observe a compelling trend: as $\Delta p$ increases, the model's performance consistently improves. This trend provides strong empirical support for our core theoretical argument that redundancy in LLMs is unevenly distributed across layers: a larger $\Delta p$ enforces stronger preservation of shallow linguistic features while allowing more aggressive compression of deep task-specific features. Such a steeper hierarchy better aligns with the intrinsic functional specialization of LLMs, thereby unlocking greater performance potential.

## A.5 COMPUTATIONAL OVERHEAD ANALYSIS

Tab. 6 shows the additional overhead introduced by LIM and ACO. The trainable parameters added by introducing LIM only come from the independent masks of each layer, totaling $2 \times L \times d_{mask}$ (where $L$ is the number of layers and $d_{mask}$ is the dimension of the masks). This order of magnitude is extremely small compared to the additional parameters introduced by PAT itself.

For example, adding LIM to the PAT baseline for Llama2-7B only increases the number of trainable parameters by 0.26M. This increment accounts for only 0.24% of the total number of parameters added by PAT itself, while slightly increasing the overall percentage of trainable parameters from 2.27% to 2.28%. The amount of additional parameters brought by ACO is also controlled at the same level. Furthermore, in our experiments, the actual training time of HA-PAT only increased by 5% to 10% compared to PAT.

Table 6: Computational Overhead Analysis

| Model | Method | Total Parameters | Trainable Parameters | Trainable Ratio |
|-------|--------|------------------|----------------------|-----------------|
| Llama2-7B | LoRA Only | 6,788,384,816 | 49,971,200 | 0.74% |
| | PAT | 6,895,153,152 | 156,737,536 | 2.27% |
| | PAT+LIM | 6,895,415,296 | 156,999,680 | 2.28% |
| | PAT+ACO | 6,897,486,912 | 159,071,296 | 2.31% |
| Llama2-13B | LoRA Only | 13,094,097,920 | 78,233,600 | 0.60% |
| | PAT | 13,260,405,760 | 244,541,440 | 1.84% |
| | PAT+LIM | 13,260,815,360 | 244,951,040 | 1.85% |
| | PAT+ACO | 13,263,231,040 | 247,366,720 | 1.87% |
| Gemma-2B | LoRA Only | 2,530,686,976 | 24,514,560 | 0.97% |
| | PAT | 2,576,230,400 | 70,057,984 | 2.72% |
| | PAT+LIM | 2,576,304,128 | 70,131,712 | 2.72% |
| | PAT+ACO | 2,579,484,624 | 73,312,208 | 2.84% |
| Gemma-7B | LoRA Only | 8,600,185,856 | 62,504,960 | 0.73% |
| | PAT | 8,705,426,432 | 167,745,536 | 1.93% |
| | PAT+LIM | 8,705,598,464 | 167,917,568 | 1.93% |
| | PAT+ACO | 8,710,381,328 | 172,700,432 | 1.98% |

## A.6 IMPLEMENTATION DETAILS

Full list of 14 downstream tasks: WSC (Levesque et al., 2012), WinoGrande (Sakaguchi et al., 2021), WIC (Pilehvar & Camacho-Collados, 2018), HellaSwag (Zellers et al., 2019), COPA (Roemmele et al., 2011), PIQA (Bisk et al., 2020), SIQA (Sap et al., 2019), ARC-C/E (Clark et al., 2018), OpenBookQA (Mihaylov et al., 2018), BOOLQ (Clark et al., 2019), MultiRC (Khashabi et al., 2018), RTE (Wang et al., 2019), and MMLU (Hendrycks et al., 2020).

To ensure fair comparison, we adopt the same rank settings for HIO and LoRA modules as in the original PAT implementation. Specifically, for the baseline using only LoRA fine-tuning, the rank of the LoRA modules is set to 64. For experiments on Gemma models, the rank of the LoRA modules is set to 20, while the rank of the HIO modules is set to 300. For LLaMA models, the LoRA rank is set to 20, and the HIO rank is set to 200.

## A.7 Overall Pipeline of HA-PAT

The overall pipeline of HA-PAT is described as Algorithm 1

---

**Algorithm 1** Overall Pipeline of HA-PAT

---

**Input**: Pre-trained LLM, Instruction-tuning dataset $\mathcal{D}_{instruct}$, Overall target pruning ratio $p$
**Parameter**: ACO low-rank $r$, Pruning ratio range $\Delta p$
**Output**: A structural pruned and fine-tuned LLM

**Initialization Phase:**
1: Insert HSMs into LLM.
2: For each layer $l \in \{1, ..., L\}$, initialize an independent trainable mask $M_l$ and a compensation matrix $D_l$.
3: Define the hierarchical pruning ratio range: $p_{min} \leftarrow p - \Delta p$, $p_{max} \leftarrow p + \Delta p$.
4: **for** each layer $l = 1$ to $L$ **do**
5:     Calculate layer-wise target pruning ratio $p_l$
6: **end for**

**Pruning-Aware Tuning Phase:**
1: **while** not converged **do**
2:     Sample a data batch and forward pass through the model with HSMs.
3:     Calculate the multi-objective loss $\mathcal{L} \leftarrow \mathcal{L}_{instruct} + \mathcal{L}_{ratio} + \mathcal{L}_{identity}$.
4:     Backward pass and optimization
5:     Update masks $\{M_l\}$ and compensation matrices $\{D_l\}$.
6: **end while**

**Finalization for Inference:**
1: Obtain final binary masks $\{M_l^*\}_{l=1}^{L}$ where values are either 0 (prune) or 1 (keep).
2: **for** each layer $l = 1$ to $L$ **do**
3:     Merge compensation matrix: $W_l' \leftarrow D_l \cdot W_l$.
4:     Physically remove the channels in $W_l'$ where the corresponding value in $M_l^*$ is 0.
5: **end for**
6: **return** Pruned and fine-tuned LLM.

---

## A.8 Detailed Main Results

We evaluate LoRA (Hu et al., 2022), LLM-Pruner (Ma et al., 2023), SliceGPT (Ashkboos et al., 2024), PAT (Liu et al., 2025), and our HA-PAT on 14 tasks. Results are shown in Tabs. 7 to 10. For experiments involving HPRS, we adopt the Linear Scheduling function, the minimum and maximum pruning ratios $p_{\min}$ and $p_{\max}$ are defined as $p - 5\%$ and $p + 5\%$, respectively. For example, when the target pruning ratio $p = 25\%$, we set $p_{\min} = 20\%$ and $p_{\max} = 30\%$.

Table 7: Zero-shot evaluation of **Llama2 7B** on 14 public datasets.

| Ratio | Method | WSC | WG | WIC | HS | COPA | PIQA | SIQA | ARC-E | ARC-C | OBQA | BOOLQ | MULTIRC | RTE | MMLU | AVG |
|---|---|---|---|---|---|---|---|---|---|---|---|---|---|---|---|---|
| 0% | LoRA-64 | 65.39 | 52.57 | 51.10 | 47.51 | 72.00 | 64.80 | 58.50 | 72.49 | 50.17 | 73.20 | 59.57 | 60.09 | 52.35 | 42.93 | 58.76 |
| 20% | LLM-Pruner | **62.50** | **52.64** | 47.34 | 34.41 | **93.00** | 65.78 | 59.88 | 71.78 | 51.19 | 68.80 | 64.50 | 60.89 | 46.93 | 39.79 | 58.53 |
| | SliceGPT | 57.69 | 52.09 | **52.04** | 34.76 | 78.00 | 65.56 | 58.24 | 65.43 | 51.19 | 67.00 | 65.05 | 60.07 | 64.26 | 37.96 | 57.81 |
| | PAT | 52.89 | 49.57 | 48.12 | **48.11** | 82.00 | **68.83** | 53.84 | 69.31 | 51.19 | 70.20 | 67.10 | 63.97 | 58.48 | 40.20 | 58.84 |
| | **Ours** | 61.58 | 51.30 | 50.94 | 46.65 | 86.00 | 68.39 | **62.34** | **74.49** | **54.56** | **71.40** | **77.26** | **68.17** | **67.87** | **41.87** | **63.06** |
| 25% | LLM-Pruner | **65.38** | **52.33** | 50.00 | 26.76 | **80.00** | 64.42 | 56.50 | 57.14 | 35.25 | 61.20 | 48.50 | 57.98 | 47.29 | 32.18 | 52.50 |
| | SliceGPT | 60.58 | 50.67 | **54.08** | 38.91 | 73.00 | 64.58 | 45.65 | 58.55 | 41.02 | 60.40 | 48.44 | 58.35 | 54.87 | 32.59 | 52.98 |
| | PAT | 47.12 | 51.07 | 50.47 | 44.23 | 72.00 | **66.00** | 52.20 | 67.02 | 47.12 | 70.00 | 65.87 | 57.01 | 63.18 | 38.02 | 56.52 |
| | **Ours** | 57.69 | 49.12 | 50.00 | **47.01** | 78.00 | 62.46 | **58.85** | **70.37** | **48.81** | **70.60** | **72.64** | **66.34** | **63.46** | **40.98** | **59.88** |
| 30% | LLM-Pruner | 50.48 | **51.07** | 52.51 | 37.04 | 75.50 | 60.88 | 44.91 | 49.82 | 35.59 | 50.10 | 63.52 | 52.71 | 51.44 | 32.77 | 50.60 |
| | SliceGPT | 48.08 | 50.95 | **53.84** | 37.82 | 67.00 | 58.87 | 47.54 | 49.65 | 40.00 | 50.20 | 64.47 | 52.92 | 58.66 | 32.79 | 50.91 |
| | PAT | 42.31 | 50.67 | 50.94 | 39.95 | **82.00** | **62.84** | 54.76 | 62.26 | 44.41 | **69.60** | 68.69 | 58.79 | **64.79** | 36.51 | 56.32 |
| | **Ours** | 59.50 | 50.51 | 53.29 | **43.43** | 76.00 | 60.99 | **56.91** | **65.08** | **48.14** | 68.40 | **73.52** | **66.67** | 61.37 | **38.58** | **58.74** |

Table 8: Zero-shot evaluation of **Llama2 13B** on 14 public datasets.

| Ratio | Method | WSC | WG | WIC | HS | COPA | PIQA | SIQA | ARC-E | ARC-C | OBQA | BOOLQ | MULTIRC | RTE | MMLU | AVG |
|---|---|---|---|---|---|---|---|---|---|---|---|---|---|---|---|---|
| 0% | LoRA-64 | 65.39 | 56.43 | 50.00 | 61.22 | 92.00 | 76.99 | 66.68 | 81.31 | 66.44 | 81.00 | 70.70 | 68.50 | 47.29 | 50.39 | 66.74 |
| 20% | LLM-Pruner | 64.42 | 53.83 | 50.47 | 62.18 | **95.00** | **75.52** | 62.79 | 77.60 | 59.66 | 76.20 | 54.07 | 66.30 | 70.40 | 45.53 | 65.28 |
| | SliceGPT | 50.96 | **56.00** | 53.68 | 57.11 | 88.50 | 71.60 | **63.84** | 77.25 | 61.19 | 78.50 | 77.46 | 72.03 | 70.76 | 43.22 | 65.86 |
| | PAT | 63.46 | 54.30 | 55.78 | 55.88 | 93.00 | 72.04 | 60.03 | **79.37** | 58.24 | **80.00** | 63.95 | 68.40 | 69.93 | 44.64 | 65.64 |
| | **Ours** | **64.65** | 52.17 | **56.47** | **65.88** | 93.00 | 70.40 | 61.06 | 78.48 | **62.03** | 79.00 | **79.12** | **76.81** | **78.95** | **47.36** | **68.96** |
| 25% | LLM-Pruner | 63.46 | 51.38 | 50.31 | 57.98 | 71.00 | 71.76 | 56.65 | 70.72 | 60.00 | 74.40 | 40.64 | 59.78 | 47.29 | 45.51 | 58.63 |
| | SliceGPT | 53.85 | **53.75** | 52.90 | 56.80 | 70.50 | 67.52 | 58.80 | 71.96 | **62.20** | 72.40 | 59.72 | 64.32 | 61.19 | 43.78 | 60.69 |
| | PAT | 63.46 | 51.46 | 50.94 | 51.91 | 88.00 | **71.82** | 58.29 | 75.84 | 56.61 | 76.40 | 62.42 | 67.45 | 68.95 | 43.51 | 63.36 |
| | **Ours** | **65.45** | 50.91 | **55.66** | 58.46 | **93.00** | 70.65 | **61.21** | **77.60** | 57.29 | 79.00 | **77.68** | **74.65** | **75.95** | **45.65** | **67.37** |
| 30% | LLM-Pruner | 51.44 | 51.10 | 50.00 | 42.89 | 63.50 | 56.28 | 50.72 | 51.06 | 40.85 | 53.80 | 65.14 | 54.92 | 50.36 | 35.87 | 51.28 |
| | SliceGPT | 64.42 | 53.63 | 50.00 | 47.31 | 70.00 | **65.53** | 51.89 | 58.47 | 51.19 | 62.10 | 58.88 | 64.99 | 47.65 | 39.55 | 56.12 |
| | PAT | 65.39 | 53.35 | 50.31 | 46.16 | 81.00 | 65.34 | 56.76 | 71.08 | 49.15 | 73.80 | 57.49 | 59.26 | 51.26 | 40.51 | 58.63 |
| | **Ours** | **66.81** | 52.49 | **56.00** | 55.16 | 88.00 | 64.32 | **57.42** | 75.13 | 58.36 | 76.85 | 78.72 | 69.91 | 75.05 | 45.46 | 65.69 |

Table 9: Zero-shot evaluation of **Gemma 2B** on 14 public datasets.

| Ratio | Method | WSC | WG | WIC | HS | COPA | PIQA | SIQA | ARC-E | ARC-C | OBQA | BOOLQ | MULTIRC | RTE | MMLU | AVG |
|---|---|---|---|---|---|---|---|---|---|---|---|---|---|---|---|---|
| 0% | LoRA-64 | 36.54 | 52.33 | 49.69 | 39.19 | 88.00 | 64.85 | 50.31 | 66.49 | 42.71 | 67.40 | 63.73 | 41.50 | 52.71 | 38.08 | 53.82 |
| 20% | LLM-Pruner | **63.46** | 50.99 | 50.00 | 34.07 | 58.50 | 56.09 | 41.48 | 46.56 | 34.58 | 56.60 | 48.24 | **63.22** | 48.92 | **31.52** | 48.87 |
| | SliceGPT | 60.58 | 50.59 | 46.08 | **34.65** | 54.00 | 52.34 | 44.63 | 45.86 | 34.92 | 49.00 | 55.44 | 60.81 | 55.24 | 30.84 | 48.21 |
| | PAT | 58.65 | **52.17** | 50.47 | 28.97 | 60.00 | **61.05** | 43.14 | 44.97 | 29.15 | 50.40 | 62.97 | 53.26 | 55.24 | 29.33 | 48.56 |
| | **Ours** | 62.54 | 50.20 | 49.84 | 33.92 | **72.00** | 60.94 | **51.59** | **61.20** | **40.00** | **66.40** | 63.85 | 54.68 | **55.96** | 30.67 | **53.84** |
| 25% | LLM-Pruner | **50.00** | 50.12 | **50.00** | 28.61 | 51.00 | 51.31 | 34.42 | 31.13 | 27.80 | 35.20 | 54.40 | 52.25 | 50.36 | 25.84 | 42.32 |
| | SliceGPT | 34.62 | **51.30** | **50.00** | 28.90 | 71.00 | 57.18 | 43.45 | 42.33 | 29.49 | 43.40 | **61.56** | 42.86 | 52.71 | 24.38 | 45.23 |
| | PAT | 34.58 | 49.67 | 48.12 | **33.31** | 71.00 | **58.32** | 48.26 | 46.03 | 32.88 | 55.40 | 58.47 | 53.40 | 50.71 | 28.96 | 47.79 |
| | **Ours** | 44.23 | 50.91 | 46.40 | 30.53 | **72.00** | 57.35 | 49.23 | **49.03** | 31.86 | **62.60** | 59.90 | 53.11 | **60.18** | 28.71 | **49.65** |
| 30% | LLM-Pruner | 36.54 | 49.57 | **50.00** | 25.05 | 55.00 | 49.51 | 32.91 | 27.87 | 21.36 | 27.50 | 62.16 | 42.80 | 52.71 | 22.93 | 39.71 |
| | SliceGPT | **63.46** | 50.43 | **50.00** | 25.06 | 45.00 | 50.44 | 32.86 | 25.40 | 27.80 | 21.20 | 37.77 | **57.20** | 47.29 | 27.12 | 40.07 |
| | PAT | 35.58 | **51.46** | 46.55 | **32.37** | 61.00 | **60.12** | 45.91 | 41.62 | 29.15 | 49.60 | 61.93 | 44.88 | 52.71 | 29.93 | 45.92 |
| | **Ours** | 39.42 | 51.30 | 49.84 | 30.89 | **70.00** | 55.66 | **46.93** | **52.38** | **30.17** | **59.80** | 66.51 | 46.43 | **57.40** | **30.82** | **49.04** |

Table 10: Zero-shot evaluation of **Gemma 7B** on 14 public datasets.

| Ratio | Method | WSC | WG | WIC | HS | COPA | PIQA | SIQA | ARC-E | ARC-C | OBQA | BOOLQ | MULTIRC | RTE | MMLU | AVG |
|---|---|---|---|---|---|---|---|---|---|---|---|---|---|---|---|---|
| 0% | LoRA-64 | 54.81 | 57.22 | 57.68 | 74.57 | 93.00 | 84.06 | 68.94 | 89.77 | 80.00 | 85.60 | 85.05 | 54.41 | 60.29 | 56.82 | 71.59 |
| 20% | LLM-Pruner | 54.09 | **55.31** | 56.31 | 62.62 | 83.25 | 72.58 | 64.29 | 81.04 | 64.83 | 80.25 | 76.01 | **60.54** | 58.84 | 46.36 | 65.45 |
| | SliceGPT | 47.60 | 55.05 | 53.06 | 62.05 | **92.50** | 74.48 | 66.68 | **85.54** | 64.92 | 81.90 | **82.28** | 58.60 | 62.64 | 45.10 | 66.60 |
| | PAT | 46.15 | 54.85 | **59.40** | 63.86 | 85.00 | 75.30 | 68.22 | 74.30 | 63.14 | **83.60** | 78.61 | 54.41 | 65.48 | 47.49 | 65.70 |
| | **Ours** | **56.73** | 54.93 | 58.61 | **65.50** | 89.00 | **75.49** | **69.31** | 80.60 | **67.46** | 79.00 | 81.77 | 58.33 | **68.85** | **50.93** | **68.32** |
| 25% | LLM-Pruner | 42.19 | 54.09 | 51.70 | 55.83 | 83.88 | 69.41 | 59.51 | 71.87 | 55.89 | 68.88 | **77.41** | 53.30 | 62.05 | 41.05 | 60.50 |
| | SliceGPT | **60.10** | **56.24** | 55.41 | **57.67** | 72.50 | 68.88 | 59.85 | 73.37 | **60.17** | 75.30 | 67.83 | **62.30** | 56.68 | **44.77** | 62.22 |
| | PAT | 37.50 | 53.28 | 58.46 | 52.96 | 84.00 | 70.99 | 64.23 | 72.19 | 54.73 | **84.20** | 70.32 | 53.17 | 63.65 | 40.39 | 61.43 |
| | **Ours** | 43.27 | 53.28 | **60.94** | 53.58 | **86.00** | **71.01** | **66.57** | **76.19** | 56.61 | 85.00 | 76.73 | 57.11 | **66.32** | 39.98 | **63.75** |
| 30% | LLM-Pruner | **52.82** | 52.26 | 50.93 | 40.23 | 64.44 | 59.95 | 46.56 | 49.69 | 40.15 | 46.44 | 57.70 | **55.24** | 54.67 | 32.89 | 50.28 |
| | SliceGPT | 42.07 | **52.47** | **51.37** | 43.47 | 72.75 | 62.00 | 49.97 | 56.61 | 43.64 | 54.75 | 72.07 | 50.70 | 57.67 | 34.44 | 53.14 |
| | PAT | 42.27 | 51.46 | 49.72 | 50.88 | 82.00 | 65.46 | 60.62 | **77.43** | **57.97** | 73.40 | 72.46 | 51.01 | **57.85** | 35.44 | 59.14 |
| | **Ours** | 43.27 | 52.28 | 50.94 | **51.58** | **85.00** | **68.01** | **61.57** | 76.19 | 56.61 | **75.00** | **76.13** | 54.11 | 56.32 | **38.98** | **60.43** |