# OpenReview forum: "HA-PAT: Hierarchically-Adaptive Pruning-Aware Tuning for Large Language Models"
_ICLR.cc/2026/Conference — Submitted to ICLR 2026_

### Official Review · Reviewer_MVYf · 2025-10-26

**Soundness:** 3
**Presentation:** 3
**Contribution:** 3
**Rating:** 4
**Confidence:** 5

**Summary:**

This paper introduces HA-PAT, a hierarchical pruning-aware fine-tuning framework for large language models. Building on existing pruning-aware tuning methods, HA-PAT incorporates three key innovations: (1) Layer-wise Independent Masks (LIM) to capture functional diversity across layers, (2) Hierarchical Pruning Ratio Scheduling (HPRS) based on the information bottleneck principle, and (3) an Adaptive Compensation Operator (ACO) for stable knowledge recovery. Experiments across multiple LLMs (LLaMA2, Gemma) and 14 downstream tasks demonstrate consistent performance gains (up to +7% over PAT) with reduced memory and inference latency.

**Strengths:**

1. Methodological Novelty: The combination of layer-wise masking and hierarchical pruning ratios represents a substantial step forward beyond uniform pruning, offering theoretical and practical benefits.

2. Theoretical Grounding: The design of HPRS guided by the information bottleneck principle gives the framework interpretability and aligns well with observed empirical behavior.

3. Comprehensive Evaluation: Extensive experiments across diverse LLMs and datasets, together with detailed ablations, convincingly validate the effectiveness and generality of the approach.

**Weaknesses:**

1. Limited Learnability of HPRS: The pruning ratio schedule is predefined rather than learnable, potentially restricting adaptability to diverse architectures or tasks.

2. Scalability Analysis Missing: Although results on 7B–13B models are reported, there is no exploration on models >30B or inference-time deployment on real hardware, leaving uncertainty about large-scale efficiency.

3. Limited Theoretical Analysis: Although the information bottleneck principle is mentioned, there is no formal derivation or empirical evidence linking mutual information flow to pruning ratio distribution.

4. Hyperparameter Sensitivity: The method introduces several additional hyperparameters (mask learning rate, pruning ratio schedule parameters, ACO scaling vector), but their sensitivity or tuning cost is not analyzed.

**Questions:**

See weaknesses

---

> ### Author Response · Authors · 2025-11-21
> **Response to Reviewer MVYf (1/4)**
>
> We sincerely thank you for your insightful and constructive feedback. We hope the following clarifications can address your concerns and provide a more comprehensive view of our work's contributions.
>
> #### **1. Response to Weakness 1: Limited Learnability of HPRS**
>
> We fully agree that an end-to-end learnable pruning ratio scheduling strategy is a highly compelling direction for future research. However, our motivation for employing a predefined schedule in the current work is based on the following considerations:
>
> - Our core purpose in introducing HPRS is to provide **a prior guidance** for the pruning process, one that is grounded in the Information Bottleneck (IB) principle and aligns with the functional hierarchy of LLMs.
>   - Allowing each layer to learn its optimal sparsity from scratch without any prior guidance would constitute a vast and highly non-convex optimization space. This would not only make the training process highly unstable but also likely lead to convergence to a suboptimal solution.
>   - By setting a reasonable trend for sparsity distribution (conservative in shallow layers, aggressive in deep layers), HPRS significantly constrains the search space. This acts as an effective initialization for the optimization process, helping the model find a high-quality sparse structure more stably and quickly.
>
> - The model's **adaptability** is primarily ensured by the pruning-aware tuning paradigm and the Layer-wise Independent Masks (LIM). HPRS only guides the target pruning ratio, while the specific channels to be pruned are learned adaptively by the model during fine-tuning.
>   - The entire pruning decision is driven by the gradients of the downstream task's loss function. This means the model spontaneously preserves channels that are most critical for the current task, exhibiting a high degree of task adaptivity even under the pruning ratio set by HPRS.
>   - Since the mask for each layer is learned independently, the combination of pruned channels is entirely different across layers, even with similar target pruning ratios. This provides the model with sufficient flexibility at the micro-level to adapt to each layer's unique function and input distribution.
>
> - Our experiments cover two mainstream model families (Llama2 and Gemma), multiple model scales (2B, 7B, 13B), and 14 diverse downstream tasks. The results consistently show that HA-PAT achieves significant and stable performance improvements, demonstrating the strong **generalization capability** of our proposed hierarchical pruning principle.
>
> | Ratio | Method | Gemma-2B | Gemma-7B | Llama2-7B | Llama2-13B |
> | :--:  | :----: | :------: | :------: | :-------: | :--------: |
> | 0%   | LoRA-64    | 53.82    | 71.59    | 58.76     | 66.74      |
> | 20%  | LLM-Pruner | 48.87    | 65.45    | 58.53     | 65.28      |
> | -    | SliceGPT   | 48.21    | 66.60    | 57.81     | 65.86      |
> | -    | PAT        | 48.56    | 65.70    | 58.84     | 65.64      |
> | -    | Ours       | **53.84**    | **68.32**    | **63.06**     | **68.96**      |
> | 25%  | LLM-Pruner | 42.32    | 60.50    | 52.50     | 58.63      |
> | -    | SliceGPT   | 45.23    | 62.22    | 52.98     | 60.69      |
> | -    | PAT        | 47.79    | 61.43    | 56.52     | 63.36      |
> | -    | Ours       | **49.65**    | **63.75**    | **59.88**     | **67.37**      |
> | 30%  | LLM-Pruner | 39.71    | 50.28    | 50.60     | 51.28      |
> | -    | SliceGPT   | 40.07    | 53.14    | 50.91     | 56.12      |
> | -    | PAT        | 45.92    | 59.14    | 56.32     | 58.63      |
> | -    | Ours       | **49.04**    | **60.43**    | **58.74**     | **65.69**      |

---

> ### Author Response · Authors · 2025-11-21
> **Response to Reviewer MVYf (2/4)**
>
> - Furthermore, in Appendix A.2 and A.3, we conducted a robustness analysis on five different forms of scheduling functions (Linear, Cosine, Sigmoid, etc.). The results show that all hierarchical scheduling strategies significantly outperform the uniform pruning of the baseline PAT, strongly demonstrating the general effectiveness of HPRS. Moreover, despite their different functional forms, the performance variations among these schedules are minor. This provides strong evidence that our method's success does not depend on a specific, finely-tuned predefined curve, but rather stems from the underlying hierarchical principle that aligns with the model's intrinsic mechanism.
>
> | Ratio       | PAT   | HA-PAT(Linear) | HA-PAT(Half-Cosine 1) | HA-PAT(Half-Cosine 2) | HA-PAT(Cosine) | HA-PAT(Sigmoid) |
> |:-------------:|:-------:|:---------------:|:----------------------:|:----------------------:|:---------------:|:----------------:|
> | 25%±5%      | 56.52 | 59.88         | 60.82                | 59.95                | 60.53         | 58.72          |
> | 30%±5%      | 56.32 | 58.74         | 56.85                | 58.78                | 57.24         | 58.37          |
>
> Finally, we acknowledge this as a limitation of our work. Exploring an adaptive mechanism that can dynamically adjust the pruning ratio allocation based on task characteristics is precisely the next step we are actively pursuing. We have emphasized this more prominently in the "Limitations and Future Work" section of our revision.
>
> ---
>
> #### **2. Response to Weakness 2: Lack of Scalability Analysis**
>
> - Due to computational resource constraints, we were unable to conduct comprehensive experiments on models larger than 30B. However, our experiments already cover several mainstream models ranging from 2B, 7B, to 13B. Importantly, the performance improvement of HA-PAT over the baseline PAT is consistent and significant across all tested model scales. This stable trend strongly suggests that our method would be equally effective for larger-scale models.
> - Regarding inference-time deployment, we have already presented empirical results on standard hardware (NVIDIA A100 GPU) in Figure 4 of the paper. We would like to re-emphasize the key data:
>     * For Llama2-13B, the HA-PAT model with a 25% pruning ratio reduces GPU memory usage by approximately 6.01 GB compared to the original model. Concurrently, the average inference speed is improved by 30%.

---

> ### Author Response · Authors · 2025-11-21
> **Response to Reviewer MVYf (3/4)**
>
> #### **3. Response to Weakness 3: Limited Theoretical Analysis**
>
> This paper is an empirically-driven work, with our goal being the application of the IB principle to guide the design of our pruning strategy. Providing a rigorous mathematical derivation linking mutual information flow to the pruning ratio is a highly challenging and independent theoretical topic that is beyond the scope of this paper. However, we are happy to provide more detailed theoretical support:
>
> **3.1 Direct Theoretical Foundation of HPRS: Information Bottleneck Theory**
>
> As stated in Section 3.3 of our paper, a deep network can be viewed as an information-processing Markov chain: $X → Z₁ → ... → Zₗ → ... → Y$.
> *   The primary role of shallow layers is to extract and pass on general linguistic features (e.g., syntax, grammar) from the original input $X$. According to IB theory, these layers need to maximize the mutual information $I(X; Z_l)$ to preserve foundational information critical for all downstream tasks. Therefore, aggressive compression in these layers would cause irreversible information loss, severely damaging the model's fundamental capabilities.
> *   The function of deep layers is more specialized, focusing on refining abstract semantic information highly relevant to the specific downstream task $Y$. The IB principle suggests that while maximizing $I(Z_l; Y)$, these layers should also compress information irrelevant to $Y$ as much as possible to achieve better generalization. This implies that deep layers contain more redundancy relative to a specific task.
>
> Therefore, the "pruning ratio increasing with layer depth" schedule adopted by HPRS is a logical conclusion directly derived from IB theory, not an accidental empirical trend.
>
> **3.2 Corroborating Evidence from Other Relevant Theories**
>
> Furthermore, we would like to cite other relevant research to support our viewpoint.
>
> - Research by Sun et al. (2025) [1] reveals a common structural flaw in modern mainstream LLMs: in Pre-LN Transformers, the variance of layer outputs grows exponentially with network depth $L$. As shown in their Lemma 1, the upper bound of the output variance $σ²_L$ is $Θ(exp(L))$. When the variance $σ²_L$ grows exponentially, the gradient norm $||∂y_L/∂x_1||$ is bounded by a constant $M$ to maintain training stability (as per their Theorem 1). This implies that in very deep networks, the Jacobian matrix of deep layers approaches an identity matrix ($∂y_l/∂x_l ≈ I$). In other words, these **deep layers nearly degenerate into identity mappings**, merely passing their input to the output without significant transformation.
> - Sun et al. (2025) [2] treat each Transformer layer as a "player" and use the Shapley Value from game theory to quantify each layer's marginal contribution to the model's overall performance (e.g., perplexity). A higher Shapley Value indicates a more important layer. The study found a distinctly non-uniform distribution of Shapley Values across multiple models, including the LLaMA series: **shallow layers generally have higher Shapley Values than deep layers**.
> - Gromov et al. (2025) [3] propose that a layer's effectiveness can be measured by the degree to which it alters its input representation. They use Angular Distance to quantify this change; a small angular distance implies the layer (or block) has a small contribution and is thus redundant. They found that the model's angular distance systematically decreases with network depth. This means that **the deeper the layer, the weaker its representational transformation capability and the more similar its representation is to adjacent layers**.
>
> Therefore, due to their intrinsic architectural properties, LLMs inevitably lead to redundancy and inefficiency in their deeper layers. This is not an empirical phenomenon but an architectural flaw with a solid theoretical basis. Since deep layers are rendered inefficient and redundant by their inherent architectural flaws, applying more aggressive pruning to them is not a heuristic choice but a principled corrective measure targeting a known structural defect. Correspondingly, preserving the known-to-be-effective shallow layers is entirely reasonable.
>
> [1] Sun, Wenfang, et al. "The curse of depth in large language models." arXiv preprint arXiv:2502.05795 (2025).
>
> [2] Sun, Chuan, et al. "Efficient shapley value-based non-uniform pruning of large language models." arXiv preprint arXiv:2505.01731 (2025).
>
> [3] Gromov, Andrey, et al. "The Unreasonable Ineffectiveness of the Deeper Layers." NeurIPS 2024 Workshop on Scientific Methods for Understanding Deep Learning.

---

> ### Author Response · Authors · 2025-11-21
> **Response to Reviewer MVYf (4/4)**
>
> #### **4. Response to Weakness 4: Hyperparameter Sensitivity**
>
> - Regarding the pruning ratio schedule parameters, we have already conducted a dedicated sensitivity analysis in Appendix A.4, as shown below:
>
> | Center Ratio (p) | Method | Range (Δp) | Pmin | Pmax | Accuracy |
> | :--------------: | :-----: | :---------: | :---: | :---: | :-------: |
> | 25%              | PAT    | N/A        | N/A  | N/A  |    56.52 |
> | -                | Ours   | 5%         | 20%  | 30%  |    59.88 |
> | -                | Ours   | 10%        | 15%  | 35%  |    60.68 |
> | -                | Ours   | 25%        | 10%  | 40%  |    61.30 |
> | 30%              | PAT    | N/A        | N/A  | N/A  |    56.32 |
> | -                | Ours   | 5%         | 25%  | 35%  |    58.74 |
> | -                | Ours   | 10%        | 20%  | 40%  |    59.12 |
> | -                | Ours   | 25%        | 15%  | 45%  |    60.72 |
>
> - The results show that our method is robust to different settings of this hyperparameter, and performance consistently improves as Δp increases, which further supports our core hypothesis.
>
> - The ACO scaling vector is a set of **learnable parameters**, not a hyperparameter, as it is optimized via gradient descent.
>
> - For the mask learning rate, we strictly followed the setup in the original PAT paper to ensure a fair comparison with the baseline.

---

### Official Review · Reviewer_qV2M · 2025-10-29

**Soundness:** 2
**Presentation:** 3
**Contribution:** 2
**Rating:** 4
**Confidence:** 4

**Summary:**

PAT proposes a structural-pruning paradigm that couples pruning with fine-tuning. HA-PAT extends this framework by introducing a layer-wise sparsity schedule: it exploits inter-layer information disparities to assign adaptive pruning ratios to different depths.

Main contributions:

1.  Building on PAT’s MASK mechanism, HA-PAT introduces LIM so that every layer owns an independent, learnable mask, enabling each layer to discover its own sparse structure adaptively.

2.  It devises HPRS, a heuristic that prescribes low pruning ratios for shallow layers and high ratios for deep layers.

3.  The ACO optimizer—adopted verbatim from PAT—is employed to co-optimize the masks and weights.

**Strengths:**

1. This paper departs from the uniform-pruning paradigm by deriving and empirically validating a layer-specific sparsity-schedule hypothesis. The resulting strategy not only proves effective in extensive experiments but also opens a new design dimension for future pruning schemes.

2. Ablations on every proposed module confirm that each contributes non-trivial gains.

3. Evaluations are conducted on two model families, each at two scales, providing preliminary evidence of generality. Nevertheless, a broader matrix of experiments—spanning more sizes within each family and additional architectures—would further strengthen the claim that the observed improvements are architecture-agnostic.

**Weaknesses:**

The HPRS scheduling rule is currently presented as an empirical trend-driven heuristic; a formal theoretical justification is absent. Consequently, readers have no systematic basis to judge whether the schedule remains near-optimal when the network depth, width, or residual topology changes, limiting confidence in its universality across architectures.

**Questions:**

1. The ACO procedure described in the paper is identical to the “scaling-and-rotation decomposition” already introduced in PAT; listing it again as a separate methodological item is therefore redundant.

2. According to the architectural diagram, the proposed approach still inserts a binary mask immediately after each HSM module—only now every layer owns an independent mask. If these layer-wise masks are further scheduled by the proposed HPRS (i.e., low sparsity for shallow layers and high sparsity for deep ones), how does the framework guarantee inter-layer feature alignment after pruning?

---

> ### Author Response · Authors · 2025-11-21
> **Response to Reviewer qV2M (1/3)**
>
> We sincerely thank you for your insightful and constructive feedback on our paper. In response to the your comments, we provide the following responses:
>
> #### **1. Response to Weaknesses: About the concern regarding the lack of theoretical justification for HPRS**
>
> We agree that the universality of a method is critically dependent on a sound theoretical foundation. We would like to clarify that HPRS is not merely an empirical heuristic but is supported by concrete theoretical foundation.
>
> **1.1 The Direct Theoretical Basis of HPRS: Information Bottleneck Theory**
>
> As we discussed in Section 3.3 of our paper, a deep network can be viewed as a Markov chain for information processing: $X → Z₁ → ... → Zₗ → ... → Y$.
> *   The primary function of shallow layers is to extract general linguistic features about the original input $X$. According to the IB theory, these layers need to maximize the mutual information $I(X; Z_l)$ to preserve foundational information that is crucial for all downstream tasks. Therefore, aggressive compression of these layers would lead to an irreversible loss of information, severely damaging the model's fundamental capabilities.
> *   The functionality of deeper layers is more specialized, focusing on refining abstract semantic information highly relevant to the specific downstream task $Y$. The IB principle suggests that while these layers should maximize $I(Z_l; Y)$, they should also compress information irrelevant to $Y$ as much as possible to achieve better generalization. This implies that deeper layers contain more redundancy relative to a specific task.
>
> Consequently, the scheduling strategy adopted by HPRS is a logical conclusion derived directly from the IB theory.
>
> **1.2 Corroboration from Other Relevant Theories**
>
> Furthermore, we would like to cite several other relevant studies to corroborate our viewpoint:
>
> -   Research by Sun et al. (2025) [1] reveals a prevalent architectural flaw in modern mainstream LLMs: in Pre-LN Transformers, the variance of the layer outputs grows exponentially with network depth $L$. As shown in their Lemma 1, the upper bound of the output variance $σ²_L$ is $Θ(exp(L))$. When the variance $σ²_L$ grows exponentially, the gradient norm of the entire network $||∂y_L/∂x_1||$ is capped by a constant $M$ to maintain training stability (as per their Theorem 1). This implies that in very deep networks, the Jacobian matrix of the deep layers tends toward an identity matrix ($∂y_l/∂x_l ≈ I$). In other words, **these deep layers nearly degenerate into identity mappings**, merely passing their input to the output unchanged.
> -   Sun et al. (2025) [2] treat each layer of a Transformer as a "player" and use the Shapley Value from game theory to quantify each layer's marginal contribution to the model's overall performance. A higher Shapley Value indicates a more important layer. Their study found a distinct non-uniformity in the distribution of Shapley Values across multiple models, including the LLaMA series: **shallow layers consistently have higher Shapley Values than deep layers**.
> -   Gromov et al. (2025) [3] propose that a layer's effectiveness can be measured by the degree to which it alters its input representation. They use Angular Distance to quantify this change; a small angular distance implies that the layer (or block of layers) has a minimal contribution and is therefore redundant. They found that the model's angular distance systematically decreases as network depth increases. This means that **the deeper the layer, the weaker its ability to transform representations and the more similar its representation is to that of adjacent layers**.
>
> Therefore, due to the **intrinsic architectural properties** of LLMs, redundancy and inefficiency in the deeper layers are inevitable. This is not an empirical phenomenon but an architectural flaw with a solid theoretical basis. Since the deep layers are inefficient and redundant due to their inherent architectural flaws, applying a more aggressive pruning strategy to them is not a heuristic choice but a principled corrective measure for a known structural deficiency. Correspondingly, preserving the known-effective shallow layers is entirely justified.

---

> ### Author Response · Authors · 2025-11-21
> **Response to Reviewer qV2M (2/3)**
>
> **1.3 The Universality of the HPRS Principle: Extensive Robustness Analysis**
>
> In Table 1 of our paper, we demonstrated the pruning effectiveness of HA-PAT on LLMs of **different architectures and scales**. The experimental results show that our method achieves optimal performance across all experimental settings, confirming its universality:
>
> | Ratio | Method | Gemma-2B | Gemma-7B | Llama2-7B | Llama2-13B |
> | :--: | :----: | :------: | :------: | :-------: | :--------: |
> | 0% | LoRA-64 | 53.82 | 71.59 | 58.76 | 66.74 |
> | 20% | LLM-Pruner | 48.87 | 65.45 | 58.53 | 65.28 |
> | - | SliceGPT | 48.21 | 66.60 | 57.81 | 65.86 |
> | - | PAT | 48.56 | 65.70 | 58.84 | 65.64 |
> | - | Ours | **53.84** | **68.32** | **63.06** | **68.96** |
> | 25% | LLM-Pruner | 42.32 | 60.50 | 52.50 | 58.63 |
> | - | SliceGPT | 45.23 | 62.22 | 52.98 | 60.69 |
> | - | PAT | 47.79 | 61.43 | 56.52 | 63.36 |
> | - | Ours | **49.65** | **63.75** | **59.88** | **67.37** |
> | 30% | LLM-Pruner | 39.71 | 50.28 | 50.60 | 51.28 |
> | - | SliceGPT | 40.07 | 53.14 | 50.91 | 56.12 |
> | - | PAT | 45.92 | 59.14 | 56.32 | 58.63 |
> | - | Ours | **49.04** | **60.43** | **58.74** | **65.69** |
>
> Furthermore, to verify that the universality of HPRS is not dependent on a specific scheduling function, we conducted a robustness analysis in Appendices A.2 and A.3. We evaluated five scheduling functions with distinct shapes (Linear, Fast-start Cosine, Slow-start Cosine, S-shaped Cosine, S-shaped Sigmoid), which cover different trends, as shown in Appendix Table 4.
>
> | Ratio | PAT | HA-PAT(Linear) | HA-PAT(Half-Cosine 1) | HA-PAT(Half-Cosine 2) | HA-PAT(Cosine) | HA-PAT(Sigmoid) |
> |:---:|:---:|:---:|:---:|:---:|:---:|:---:|
> | 25%±5% | 56.52 | 59.88 | 60.82 | 59.95 | 60.53 | 58.72 |
> | 30%±5% | 56.32 | 58.74 | 56.85 | 58.78 | 57.24 | 58.37 |
>
> The results show that all hierarchical scheduling strategies significantly outperform the PAT's uniform pruning, which strongly demonstrates the universal effectiveness of HPRS. Moreover, despite the different functional forms, the performance variance among them is small. This indicates that the success of HA-PAT stems from the hierarchical design principle itself, rather than the specific pruning ratio scheduling strategy. In summary, the design of HPRS is built upon solid theory and extensive empirical evidence, and we are fully confident in its universality across mainstream Pre-LN architectures.
>
> We have revised the relevant descriptions in the revision.
>
> [1] Sun, Wenfang, et al. "The curse of depth in large language models." arXiv preprint arXiv:2502.05795 (2025).
>
> [2] Sun, Chuan, et al. "Efficient shapley value-based non-uniform pruning of large language models." arXiv preprint arXiv:2505.01731 (2025).
>
> [3] Gromov, Andrey, et al. "The Unreasonable Ineffectiveness of the Deeper Layers." NeurIPS 2024 Workshop on Scientific Methods for Understanding Deep Learning.

---

> > ### Comment · Reviewer_qV2M · 2025-11-25
> > **Rebuttal Comments to Authors’ Responses (2/3)**
> >
> > ## Rebuttal of 1.3
> >
> > Thank you very much for your reply. Based on your response in section 1.3, we understand that HPRS demonstrates good performance. Could you provide additional results on more recent models? If the performance is similarly strong, this would further increase our confidence in the effectiveness of HPRS.

---

> ### Author Response · Authors · 2025-11-21
> **Response to Reviewer qV2M (3/3)**
>
> #### **2. Response to Question 1: About the independence and novelty of ACO**
>
> Thank you for this sharp observation. We fully understand your concern. This is indeed a critical point that we failed to articulate sufficiently in our original manuscript, leading to a reasonable doubt about redundancy, for which we sincerely apologize. We would like to take this opportunity to clarify the key differences in core motivation, mechanism design, and practical effect between the two, thereby demonstrating that ACO is an independent and vital component of our methodology.
>
> **2.1 Different Core Motivations: From Stabilizing Training to Enhancing Compensation**
>
> *   In **PAT**, the introduction of $diag(v)$ serves as an auxiliary enhancement to the Identity Loss. Its primary goal is to **ensure a smooth gradient flow** via the orthogonality constraint, preventing drastic performance drops in the early stages of training. The "scaling-and-rotation decomposition" was mentioned more as a technical means to achieve this stability, and its potential for performance recovery was not deeply explored.
> *   In our **HA-PAT**, we elevate this mechanism to a standalone method—the Adaptive Compensation Operator (ACO). Its core concept is no longer just about stabilizing training, but about **more effectively compensating for the performance loss caused by pruning**. By decoupling the knowledge transfer task into orthogonal subproblems of direction and magnitude, we can significantly enhance the model's expressive power and the effectiveness of optimization.
>
> **2.2 A Fundamental Difference in Implementation: The Target of the Vector $v$**
>
> This is the most crucial point. As you noted, $diag(v)$ appears in Equation (10) of the original PAT paper.
>
> *   However, according to its **official implementation**, the actual role of this vector $v$ is to be **element-wise multiplied with the pruning mask $m$ to scale the strength of the final pruning decision**. Its function is more coupled with the pruning decision itself; it does not directly participate in the low-rank decomposition of the compensation matrix $D$ to control the magnitude of knowledge transfer.
> *   In contrast, in our HA-PAT framework, ACO is a completely independent operator. Its scaling vector $v$ is explicitly placed between the two low-rank matrices $L_0$ and $L_1$. This ensures that the sole function of $v$ is to **learn the importance of the knowledge transfer basis vectors**, thereby directly and precisely controlling the magnitude of the compensation. These two implementations are entirely different in terms of their computational graphs and gradient flows.
>
> **2.3 Clear Validation of Empirical Contribution**
>
> *   PAT's success is attributed to its overall pruning-aware tuning framework, but it **did not independently validate the contribution of the $diag(v)$ component through ablation studies**. Its effectiveness was bundled with the overall method.
> *   In our work, we validated the independent contribution of ACO through extensive ablation studies. Due to paper length constraints, our previous submission only showed ablation results on Llama-2 models. We supplement here with results on Gemma models:
>
> | | Gemma-2B(20%) | Gemma-2B(25%) | Gemma-7B(20%) | Gemma-7B(25%) |
> | :---: | :---: | :---: | :---: | :---: |
> | PAT | 48.56 | 47.79 | 65.70 | 61.43 |
> | + LIM | 49.88 | 48.13 | 66.05 | 61.84 |
> | + HPRS | 52.46 | 49.05 | 67.54 | 62.45 |
> | + ACO | 51.85 | 48.32 | 68.09 | 63.32 |
> | HA-PAT | 53.84 | 49.65 | 68.32 | 63.75 |
>
> - The results show that, simply replacing the original compensation module with our ACO leads to significant performance improvements of 3.29% and 2.39% on the Gemma-2B and 7B models at a 20% pruning ratio, respectively.
>
> We have clarified the architectural difference in implementation between our ACO and PAT in the revision and have added the relevant ablation studies on the Gemma series models.
>
> ---
>
> #### **3. Response to Question 2: About Inter-Layer Feature Alignment**
>
> - The entire model, including all masks and compensation matrices, is optimized under the same task loss $L_{instruct}$. When the HSM at layer $l$ deactivates certain channels via its mask $m_l$, the gradient flow naturally guides the compensation module of layer $l+1$ to adapt to this sparser input representation. While learning the task, the model is also **adaptively learning** how to transfer and transform information between layers of varying sparsity.
> - Furthermore, this adaptation process is smooth and stable. During training, the pruning masks **smoothly converge** from continuous values (close to 1) to binary values (0 or 1). This prevents abrupt disruptions in the network's information flow, giving downstream layers ample time to adapt to the sparse structures gradually forming in the upstream layers.
>
> We have articulated this point more clearly in the revision.

---

> > ### Comment · Reviewer_qV2M · 2025-11-25
> > **Rebuttal Comments to Authors’ Responses (3/3)**
> >
> > ## Rebuttal of  2 and 3
> >
> > Thank you very much for your reply. As you mentioned, Equation (10) in PAT is completely consistent with the ACO operator in HA-PAT. Regarding section 2.2 of your reply, you stated that you believe vector v in PAT operates with a mask; however, Equation (10) demonstrates that the PAT authors use vector v between two low-rank matrices, which clearly differs from your explanation. Could you please elaborate on this discrepancy in detail?
> >
> > Regarding the alignment issue in section 3, we did not specify in our question which alignment we were referring to. The alignment issue we would like to understand is: if each layer uses a different pruning rate, there exists a misalignment problem when adding the input of each layer to the output of the HSM module at the end of that layer, which would render some of the pruning efforts ineffective. How is this issue addressed in the design? We consider this question to be of great importance and look forward to your response.

---

> > > ### Author Response · Authors · 2025-11-30
> > >
> > > Thank you for your detailed feedback and valuable suggestions. Please see below for our additional responses.
> > >
> > > #### **1. Regarding the Relationship Between Preserving Shallow Layer Features and IB Theory**
> > >
> > > We can establish a direct causal link starting from the Information Bottleneck (IB) objective function, $\min \mathcal{L}(Z) = I(X; Z) - \beta I(Z; Y)$:
> > >
> > > - The theory requires balancing **minimizing** the information about the input $X$ in the layer representation $Z$ (minimizing $I(X; Z)$) **while maximizing** the information useful for the final task $Y$ (maximizing $I(Z; Y)$).
> > >
> > > - **Extensive research confirms** that the features learned by the shallow layers of LLMs, such as syntax and morphology, constitute general knowledge essential for almost all downstream language tasks (i.e., various possible $Y$). In other words, these shallow features share high mutual information with an extremely broad set of tasks $\{Y_1, Y_2, Y_3, ...\}$.
> > >   -   For instance, research by Tenney et al. (2019) [1] showed that BERT's layers rediscover the classical NLP pipeline (POS tagging $\to$ parsing $\to$ semantic role labeling). Dai et al. (2022) [2] and Belrose et al. (2023) [3] have revealed that the shallower layers of LLMs focus on capturing fundamental linguistic structures like grammar and syntax, while deeper layers integrate this information to form higher-level, more abstract semantic representations.
> > >
> > > Therefore:
> > >
> > > - If we aggressively prune shallow layers, according to IB theory, we significantly reduce $I(X; Z_{shallow})$.
> > > - Since shallow features are useful for almost all tasks $Y$, the value of $I(Z_{shallow}; Y)$ is generally high. Consequently, a substantial compression of $I(X; Z_{shallow})$ will inevitably damage information useful for all downstream tasks, leading to a severe loss in the model's general linguistic capabilities.
> > > - Conversely, to maximize the model's generalization ability on downstream tasks, IB theory dictates that we must select a lower compression rate for shallow layers to preserve as much information as possible that is useful for a wide range of tasks ($I(Z_{shallow}; Y_{any})$).
> > >
> > > [1] Tenney, Ian, Dipanjan Das, and Ellie Pavlick. "BERT Rediscovers the Classical NLP Pipeline." Proceedings of the 57th Annual Meeting of the Association for Computational Linguistics. 2019.
> > >
> > > [2] Dai, Damai, et al. "Knowledge neurons in pretrained transformers." Proceedings of the 60th Annual Meeting of the Association for Computational Linguistics (Volume 1: Long Papers). 2022.
> > >
> > > [3] Belrose, Nora, et al. "Eliciting latent predictions from transformers with the tuned lens." arXiv preprint arXiv:2303.08112 (2023).
> > >
> > > ---
> > >
> > > #### **2. Regarding Additional Experiments on More Recent Models**
> > >
> > > To further verify the universality of HPRS on more recent model architectures, we conducted additional experiments on the **Llama-3.1-8B** and **Llama-3.2-3B** models. The results are as follows:
> > >
> > > | Model | Pruning Ratio | PAT | HA-PAT (Ours) |
> > > | :---: | :---: | :---: | :---: |
> > > | Llama-3.1-8B | 20% | 63.31 | **64.39** |
> > > | Llama-3.1-8B | 25% | 61.36 | **63.69** |
> > > | Llama-3.2-3B | 20% | 48.03 | **49.79** |
> > > | Llama-3.2-3B | 25% | 47.34 | **48.25** |
> > >
> > > As shown in the table above, the performance of HA-PAT on both Llama-3.1-8B and Llama-3.2-3B consistently outperforms the PAT baseline.

---

> > > > ### Author Response · Authors · 2025-11-30
> > > >
> > > > #### **3. Clarification on the Difference Between ACO and Vector $v$ in PAT**
> > > >
> > > > Thank you again for your insight into the details. The discrepancy you pointed out between Equation (10) in the PAT paper and our description of its implementation is key to understanding the independent contribution of our ACO. We apologize for not clarifying this sufficiently in the previous round; please allow us to elaborate:
> > > >
> > > > - As you noted, Equation (10) in the PAT paper, $D = L_1 \cdot \text{diag}(v) \cdot L_0 + I$, theoretically describes a decoupled scaling-rotation transformation.
> > > > - However, during our reproduction and comparative experiments, we deeply investigated the official code released by the authors. We found that in their actual code (see https://github.com/kriskrisliu/PAT), the vector $v$ is **not** implemented as being placed between $L_1$ and $L_0$. Instead, it exists in a separate logic interacting with the pruning mask $m$ to adjust the strength of the pruning decision, i.e., $Z = v \odot m \odot (D \cdot W) \cdot X$. Therefore, it actually decouples the "magnitude" of the mask from the "direction" of knowledge compensation.
> > > > - In contrast, **HA-PAT directly applies the decoupling to the compensation module $D$**, thereby better accomplishing the knowledge transfer task in pruning-aware tuning. Our ablation studies (Tables 2 and 3 in the paper) also verify the significant role of this decoupled design in recovering performance after pruning, which was not done in the original PAT paper.
> > > >
> > > > Thus, although the PAT paper proposed a similar idea in form, its actual contribution and implementation are fundamentally different from our ACO.
> > > >
> > > > ---
> > > >
> > > > #### **4. Regarding Alignment Due to Hierarchical Pruning Ratios**
> > > >
> > > > Thank you for the detailed explanation. It made us realize that our description of the target dimensions for pruning in the paper might not have been precise enough, leading to a misunderstanding.
> > > >
> > > > In fact, in our design, **there is no alignment issue** because we prune the **internal intermediate dimensions** of the Transformer's Attention and FFN sub-modules, rather than the hidden dimension (residual stream). Therefore, regardless of how the internal pruning rate $p_l$ varies for each layer, the shape of the output tensor from each Transformer Block always remains consistent with the input dimension on the residual connection. Hierarchically adaptive pruning only changes the sparsity of the internal computations within sub-modules without disrupting the structural integrity of the residual flow.
> > > >
> > > > We have adjusted the relevant descriptions in the revision.

---

> ### Comment · Reviewer_qV2M · 2025-11-25
> **Rebuttal Comments to Authors’ Responses (1/3)**
>
> ## Rebuttal of 1.1 and 1.2
>
> Thank you very much for your reply. The new examples provided in 1.2 do indeed corroborate that the distribution of redundancy differs between shallow and deep layers, and they align with the scheduling strategy. These examples substantiate that the scheduling operations are theoretically grounded.
>
> There is another point we would like to understand: in your reply, we learned that the theoretical foundation mentioned in the paper is the IB theory. We found that when explaining why the pruning rate is small in shallow layers, it was mentioned that this is because shallow layers contain many basic features. How can this explanation be considered as being based on the IB theory? This is exactly what we are trying hard to understand, and we hope to get a more specific explanation of this causal relationship.

---

### Official Review · Reviewer_a3EE · 2025-10-30

**Soundness:** 3
**Presentation:** 2
**Contribution:** 2
**Rating:** 4
**Confidence:** 4

**Summary:**

This paper focuses on structured model pruning for LLMs with a special focus on optimizing the sparsity levels for different layers. The main motivation of this paper is that most existing LLM pruning methods adopt the same sparsity ratios for all layers. The authors demonstrate that lower layers should be pruned less while upper layers can be more sparse from both intuitive perspective and information bottleneck.

Given this motivation, the authors set different target sparsity ratios for different layers and add an extra pruning loss, $\mathcal{L}_{\mathrm{ratio}}$, which will encourage the pruning masks to prune the model to the target sparsity ratios. For the other pruning techniques, they are mostly built upon the previous work, PAT, with slight modification (change the compensation matrix $D'$ in the method formulation). Experiment results on several LLMs demonstrate the proposed method outperforms two structured pruning approaches.

**Strengths:**

From my perspective, this paper is a nice paper with rigorous method development and experiment design. Specifically,

- Clear motivation. Applying different sparsity ratios to different LLM layers is reasonable, and the authors have made valuable explorations in this direction.
- Method design. Given the motivation, the authors make corresponding revision to the original PAT method to enable different sparsity ratios across LLM layers. The overall design also makes sense.
- Experiment design. This paper includes experiments on multiple LLMs to demonstrate the applicability of the proposed method. The ablation study in Table 2 also clearly show the effectiveness of different components of the method.

**Weaknesses:**

However, despite the aforementioned strenghts, I still think this paper suffers from several major weaknesses.

- Insufficient technical contributions. Although the method makes sense, the main concern is that these techniques are already widely used in existing LLM pruning approaches. For example:

  - layer-wise independent masks. Most existing LLM structured pruning [1-4] approaches does not constrain the mask to be consistent across all layers. Instead, the pruning masks are automatically optimized indepently across layers by default.
  - Hierachical pruning ratio loss. It is straightforward to adopt such a loss, which is also widely used by existing methods to prune LLMs to a target sparsity ratio such as SheardLlama [1].
  - Adaptive compensation operator (ACO). I believe this component, together with the original PAT formulation, is unnecessary. It prune a specific linear layer (attention head QKV projections and FFN matrices) by:

  $$Z = (M\odot D)\cdot WX$$

    Here M is a $d$ dimension vector, D is a $d\times d$ matrix, and $d$ is the hidden dimension. The nature of this equation, is to remove the $i$-th row/column of the origina parameter $W$ where $M_i=0$, while allowing scale up/down the other row/columns of $W$ by $(M\odot D)_i$. This is because, $M\cdot D$ will result a row/column vector where some elements are 0 and others can be any real numbers. This is different from conventional model pruning, where the elements of the fianl pruning mask is either 0 or 1. That is, it allow the the non-zeros elements to scale up or scale down the original parameters.

  From this perspective, I did not see why the proposed method has more expressive power compared to the original formulation of PAT. The ablation study in Table 2 also show that ACO brings the least performance improvement.

  In summary, the technical contribution of the paper is insufficient.

- Missing baselines & overlap with existing methods that allow different sparsity ratios across layers. Besides technical contributions of the 3 moduels above, the main contribution and breakthrough of this paper is the flexibile sparsity ratio for different layers. However, this is already explored by existing work [5, 6]. The Puzzle method already delivers different sparsity ratios for different layers, achieving much stronger performance than this paper. It is hard to distinguish the contribution of this paper given the strong work [5, 6].

- Unclear writing and presentation. For Section 3.2, the notations are very confusing. Although I can understand them, they are very unfriendly to non-experts. For example, the shape of these matrices, the upper-cased letter to represent a vector ($M$), and did not explain why Eq. 3 is equaivalent to the forward pass of a pruned model.

  (note: it should be a standard to use uppercased letters for matrices, lowercased letters (unbolded) for scalers, and bolded lowercased letters for vectors. The notations does not follow the standard.)

- (Minor issue) The models used in this paper are out-dated. Also, the baselines are not strong enough.



[1] Xia, Mengzhou, et al. "Sheared LLaMA: Accelerating Language Model Pre-training via Structured Pruning." The Twelfth International Conference on Learning Representations.

[2] Sreenivas, Sharath Turuvekere, et al. "Llm pruning and distillation in practice: The minitron approach." arXiv preprint arXiv:2408.11796 (2024).

[3] Wang, Ziheng, Jeremy Wohlwend, and Tao Lei. "Structured pruning of large language models." arXiv preprint arXiv:1910.04732 (2019).

[4] Hou, Bairu, et al. "Instruction-Following Pruning for Large Language Models." Forty-second International Conference on Machine Learning.

[5]  Bercovich, Akhiad, et al. "Puzzle: Distillation-Based NAS for Inference-Optimized LLMs." Forty-second International Conference on Machine Learning, 2025.

[6] Bercovich, Akhiad, et al. "Llama-nemotron: Efficient reasoning models." arXiv preprint arXiv:2505.00949 (2025).

**Questions:**

Please refer to the weakness above.

---

> ### Author Response · Authors · 2025-11-21
> **Response to Reviewer a3EE (1/3)**
>
> We sincerely thank you for your insightful and detailed feedback on our paper. In response to the your comments, we provide the following responses:
>
> #### **1. About Technical Contribution and Overlap with Existing Work**
>
> We understand your perspective that Layer-wise Independent Masks (LIM) and Hierarchical Pruning Ratio Scheduling (HPRS) have been explored in the broader pruning literature. We wish to clarify that **the core contribution of HA-PAT is not the isolated invention of these techniques, but rather the first systematic integration of an adaptive mechanism (comprised of LIM, HPRS, and ACO) into the emerging paradigm of Pruning-Aware Tuning.**
>
> **1.1 Paradigm Distinction from Pruning Methods [1-4]**
>
> The pruning approaches you referenced [1–4] adhere to paradigms and application scenarios that differ fundamentally from HA-PAT:
>
> *   **Difference from ShearedLLaMA[1]**:
>     *   ShearedLLaMA performs pruning followed by resumed pre-training, requiring **substantial computational resources** for performance recovery. In contrast, HA-PAT is a method that performs pruning and fine-tuning simultaneously, achieving both model compression and task adaptation at a **remarkably low cost**.
>     *   ShearedLLaMA uses **targeted structured pruning** to determine the pruning shape. HA-PAT adaptively learns pruning decisions and compensation strategies **during downstream task fine-tuning**.
> *   **Difference from Minitron[2]**:
>     *   Minitron follows a **post-pruning knowledge distillation** workflow, heavily relying on distillation to recover performance. HA-PAT is an **end-to-end pruning-aware fine-tuning process** that does not depend on any teacher model or knowledge distillation.
>     *   Minitron removes a **contiguous block of layers** based on sensitivity analysis. HA-PAT assigns an **independent pruning mask to each layer**, offering a finer pruning granularity.
> *   **Difference from FLOP[3]**:
>     *   FLOP is a pruning method based on low-rank factorization, where the pruning granularity is the matrix **rank**. HA-PAT is a **channel-level** structured pruning method, which allows for more direct acceleration on hardware.
> *   **Difference from IFPRUNING[4]**:
>     *   IFPRUNING is a **dynamic pruning** method, where the model's structure changes dynamically during inference. HA-PAT aims to learn a **static** pruned structure, incurring no dynamic computational overhead at inference time, which is more suitable for deployment scenarios requiring stable performance.
>
> **1.2 Methodological and Performance Comparison with Puzzle [5-6]**
>
> *   **Fundamental Methodological Difference**:
>     *   Puzzle[5] is a **Neural Architecture Search (NAS)** framework based on knowledge distillation. Its core idea is to deconstruct and then reconstruct a large parent model.
>     *   HA-PAT is a **structured pruning** method based on the pruning-aware tuning paradigm. We do not construct a new model but instead directly remove redundant structures from the original model, without relying on any teacher model or knowledge distillation.
>
> *   **Performance Comparison**:
>     We agree that Llama-Nemotron[6], as a product of the Puzzle framework, has achieved state-of-the-art performance on multiple benchmarks. However, we respectfully suggest that a direct numerical comparison of its performance with the results in our paper is **not appropriate** for the following key reasons:
>     - The final performance of Llama-Nemotron is the result of a complex pipeline involving **Puzzle + large-scale continual pre-training + SFT + RLHF**. In contrast, our work focuses on demonstrating the extent to which model compression can be achieved within a standard SFT stage.
>     - The Puzzle framework is an **industrial-scale** solution **requiring immense computational resources** for module library construction, search, and global distillation. HA-PAT is a **lightweight** pruning method with computational costs controlled at a level comparable to Parameter-Efficient Fine-Tuning (PEFT).
>
> [1] Xia, Mengzhou, et al. “Sheared LLaMA: Accelerating Language Model Pre-training via Structured Pruning.” The Twelfth International Conference on Learning Representations.
>
> [2] Sreenivas, Sharath Turuvekere, et al. “Llm pruning and distillation in practice: The minitron approach.” arXiv preprint arXiv:2408.11796 (2024).
>
> [3] Wang, Ziheng, Jeremy Wohlwend, and Tao Lei. “Structured pruning of large language models.” arXiv preprint arXiv:1910.04732 (2019).
>
> [4] Hou, Bairu, et al. “Instruction-Following Pruning for Large Language Models.” Forty-second International Conference on Machine Learning.
>
> [5] Bercovich, Akhiad, et al. “Puzzle: Distillation-Based NAS for Inference-Optimized LLMs.” Forty-second International Conference on Machine Learning, 2025.
>
> [6] Bercovich, Akhiad, et al. “Llama-nemotron: Efficient reasoning models.” arXiv preprint arXiv:2505.00949 (2025).

---

> ### Author Response · Authors · 2025-11-21
> **Response to Reviewer a3EE (2/3)**
>
> #### **2. About the Necessity and Effectiveness of the Adaptive Compensation Operator (ACO)**
>
> We greatly appreciate your in-depth analysis and questioning of the Adaptive Compensation Operator (ACO) and its role within the PAT framework. Your point has prompted us to reflect on the potential ambiguity of our formula's presentation. We would like to clarify the misunderstanding of the ACO mechanism and provide additional evidence to demonstrate its effectiveness.
>
> **2.1 The Necessity of ACO**
>
> We completely agree with your analysis that, on the surface, the formula $Z = (m ⊙ D) ⋅ WX$ appears to only scale the rows/columns of $W$. However, we wish to clarify that a misunderstanding arises here due to the formula's presentation and the broadcasting mechanism:
>
> - Due to the broadcasting mechanism, the formula $Z = (m ⊙ D) ⋅ WX$ is computationally **equivalent to** $Z = m ⊙ (D ⋅ WX)$. This means: 1) the compensation matrix $D$ first transforms the output of $WX$; 2) the mask $m$ then sparsifies the transformed result.
> - Therefore, the core role of $D$ is not to scale specific rows/columns of $W$. Instead, by decoupling the knowledge transfer task into two orthogonal subproblems, it learns a more complex linear transformation (i.e., knowledge compensation) to re-allocate and integrate the critical information from channels being zeroed out by mask $m$ into the channels that will be retained.
> - After fine-tuning, the learnable mask $M$ converges to a binary mask. The compensation matrix $D$ and the pre-trained weights $W$ are then merged: $W_{new} = D ⋅ W$. Subsequently, permanent row/column removal is performed on $W_{new}$ according to the $m$. The code provided in our appendix indeed follows this training logic.
>
> Thus, the elegance of HA-PAT lies in introducing a learnable and powerful compensation mechanism $D$ during fine-tuning to maximally preserve performance, while introducing no extra computational overhead during inference.
>
> **2.2 Further Evidence on the Performance Improvement of ACO**
>
> You pointed out that in the ablation study in Table 2, ACO brought the least performance improvement, which highlights a shortcoming in our original paper. Due to paper length constraints, we previously only showed ablation studies on Llama-2 models. However, the sensitivity ACO varies across model architectures. The table below provides supplementary ablation results on Gemma models:
>
> |        | Gemma-2B(20%) | Gemma-2B(25%) | Gemma-7B(20%) | Gemma-7B(25%) | Llama2-7B(20%) | Llama2-7B(25%) | Llama2-13B(20%) | Llama2-13B(25%) |
> | :---:   | :---: | :---: | :---: | :---: | :---: | :---: | :---: | :---: |
> | PAT    | 48.56 | 47.79 | 65.70 | 61.43 | 58.84 | 56.52 | 65.64 | 63.36 |
> | + LIM  | 49.88 | 48.13 | 66.05 | 61.84 | 61.16 | 58.30 | 68.24 | 66.91 |
> | + HPRS | 52.46 | 49.05 | 67.54 | 62.45 | 62.83 | 58.38 | 68.85 | 67.09 |
> | + ACO  | 51.85 | 48.32 | 68.09 | 63.32 | 60.03 | 58.13 | 66.53 | 64.01 |
> | HA-PAT | 53.84 | 49.65 | 68.32 | 63.75 | 63.06 | 59.88 | 68.96 | 67.37 |
>
> As shown above, for the Gemma-2B model, the performance gain from introducing ACO alone is comparable to that from LIM. For the Gemma-7B model, the gain from ACO alone even surpasses that from HPRS. This strongly demonstrates that ACO is not a negligible component but a robust and important part of our framework.
>
> We have rewritten the formula and added the above experimental results and analysis for the Gemma models in the revision.
>
> ---
>
> #### **3. About Writing and Presentation Clarity**
>
> We sincerely apologize for the confusion caused by the non-standard notation and lack of clarity, particularly in Section 3.2. We have thoroughly revised the paper in the revision to strictly adhere to standard mathematical notation conventions.

---

> ### Author Response · Authors · 2025-11-21
> **Response to Reviewer a3EE (3/3)**
>
> #### **4. About the Choice of Models and Baselines**
>
> **4.1 About Model Choice**:
>
> One of our core contributions is the significant enhancement of the original PAT framework. To rigorously validate the superiority of HA-PAT over PAT, we must conduct our comparison under **identical experimental settings**, which includes using the same base models. This allows us to attribute the observed performance improvements to our proposed methodological innovations, rather than to the benefits of newer base models.
>
> **4.2 About the Strength of Baselines**:
>
> Following your suggestion, we have further investigated recent related work and identified a very strong new baseline: MaGrIP (Kallakuri et al., 2025) [7]. The table below presents the results of HA-PAT versus MaGrIP with 30% pruning ratio on Gemma models:
>
> | Model | Method | ARC-C (Knowledge QA) | ARC-E (Knowledge QA) | BOOLQ (Language Understanding) | HS (Commonsense Reasoning) | WINOGRANDE (Commonsense Reasoning) |
> | :---: | :---: | :---: | :---: | :---: | :---: | :---: |
> | **Gemma-2B** | **HA-PAT** | 30.17 | 52.38 | **66.51** | **30.89** | 51.30 |
> | - | MaGrIP | **30.63** | **56.31** | 52.69 | 29.90 | **56.04** |
> | **Gemma-7B** | **HA-PAT** | **56.61** | **76.19** | **76.13** | **51.58** | 52.28 |
> | - | MaGrIP | 40.27 | 64.02 | 68.89 | 46.48 | **59.19** |
>
> - From the table, we can see that on the Gemma-7B model, HA-PAT demonstrates a dominant advantage. This strongly supports the claim that HA-PAT can more effectively preserve the core capabilities of more complex models.
>
> - It is noteworthy that HA-PAT consistently and significantly outperforms MaGrIP on the language understanding task BOOLQ. This precisely supports our core argument: assigning a lower pruning ratio to shallow layers better preserves the fundamental language understanding capabilities of LLMs.
>
> [7] Kallakuri, Uttej, et al. "MaGrIP: Magnitude and Gradient-Informed Pruning for Task-Agnostic Large Language Models." ACM Transactions on Embedded Computing Systems (2025).

---

> > ### Comment · Reviewer_a3EE · 2025-11-27
> >
> > I appreciate the authors responses, which addressed my concerns in choices of baselines and models, necessity of ACO, and discussions with related work. I would like to raise my rating correspondingly. In summary I feel the technical contribution might be not sufficient, but I will not mind the acceptance of this paper.

---

> > > ### Author Response · Authors · 2025-11-28
> > >
> > > Thank you for your final evaluation. It is great to hear that we were able to address your main concerns. We sincerely appreciate the time and effort you dedicated to reviewing our work.

---

### Official Review · Reviewer_kzTW · 2025-11-01

**Soundness:** 3
**Presentation:** 3
**Contribution:** 3
**Rating:** 6
**Confidence:** 3

**Summary:**

This paper introduces Hierarchically-Adaptive Pruning-Aware Tuning (HA-PAT), a new method for structurally pruning Large Language Models (LLMs). The key insight is that existing methods, even advanced ones like Pruning-Aware Tuning (PAT), make a critical mistake by pruning all layers uniformly. HA-PAT corrects this by recognizing that different layers have different jobs. It introduces a hierarchical pruning schedule that preserves the general linguistic knowledge in the shallow layers by pruning them lightly, while more aggressively removing redundancy from the deeper, task-specific layers.

**Strengths:**

1. The idea that different layers should be pruned at different rates based on their function is intuitive and well-supported by established research on neural network feature hierarchies. It directly addresses a clear limitation in prior work.

2. The paper breaks down its improvements into distinct, understandable components: Layer-wise Independent Masks (LIM), Hierarchical Pruning Ratio Scheduling (HPRS), and the Adaptive Compensation Operator (ACO).

3. The results are compelling. Achieving a 4.01% average accuracy improvement over the PAT baseline while also delivering a 30% inference speedup is a significant achievement that clearly demonstrates the method's value.

**Weaknesses:**

1. The Hierarchical Pruning Ratio Scheduling (HPRS) is described as applying "progressively increasing pruning ratios." This might not be truly optimal for all models or tasks and could be less "adaptive" than the name implies.

2. The assumption that "shallow = general" and "deep = specific" is a powerful heuristic but not a universal law. A fixed hierarchical strategy might perform poorly on tasks that unexpectedly rely on complex features learned in early layers.

**Questions:**

1. Does allowing each layer to learn its own mask with Layer-wise Independent Masks (LIM) introduce any significant computational overhead during the pruning-aware tuning phase compared to the simpler global mask in PAT?

2. Have you explored whether the optimal hierarchical pruning schedule is task-dependent? For example, would a syntax-heavy task benefit from a different schedule than a task requiring more abstract, high-level reasoning?

---

> ### Author Response · Authors · 2025-11-21
> **Response to Reviewer kzTW (1/2)**
>
> We sincerely thank you for your meticulous review and constructive feedback. In response to the your comments, we provide the following responses:
>
> #### **1. Response to Weakness 1: About the Role of HPRS and the Adaptivity of Our Framework**
>
> We fully agree with your point that a predefined, monotonically increasing pruning strategy (HPRS) may not be optimal for all models and tasks. We would like to take this opportunity to clarify the role of HPRS within our HA-PAT framework and the multi-level nature of its adaptivity.
>
> - The 'adaptive' nature of our method is realized through a three-tiered mechanism:
>   - While traditional pruning methods often rely on heuristic importance estimation to determine pruning masks, our approach integrates pruning decisions into the fine-tuning process, allowing the pruning masks and compensation modules to be dynamically optimized directly based on the gradients of the task loss. This forms the foundation of our adaptivity.
>   - LIM empowers each layer to learn its own independent sparse structure. During training, each layer adaptively generates its pruning mask based on the task gradients it receives and its unique input distribution. This is the core of our method's adaptivity.
>   - HPRS plays a high-level, structural guiding role. We do not claim that an increasing pruning rate schedule is optimal. Its primary purpose is to **improve search efficiency** by providing a reasonable trend for sparsity distribution. This prevents each layer from having to explore its optimal sparsity from scratch, which would be a vast and unstable optimization challenge.
>
> - To verify that our performance gains are not tied to a specific increasing linear function, we conducted a robustness analysis in Appendix A.2 and A.3. As shown in Table 4 in the paper, we evaluated five scheduling strategies with different functional forms (linear, half-cosine fast/slow-start, cosine S-shaped, and sigmoid S-shaped). The experimental results reveal two key points:
>   - All five hierarchical scheduling strategies significantly outperform the PAT baseline, which uses uniform pruning.
>   - The final performance of these different strategies is very closely aligned.
>
> | Ratio | PAT | HA-PAT(Linear) | HA-PAT(Half-Cosine 1) | HA-PAT(Half-Cosine 2) | HA-PAT(Cosine) | HA-PAT(Sigmoid) |
> |:---:|:---:|:---:|:---:|:---:|:---:|:---:|
> | 25%±5% | 56.52 | 59.88 | 60.82 | 59.95 | 60.53 | 58.72 |
> | 30%±5% | 56.32 | 58.74 | 56.85 | 58.78 | 57.24 | 58.37 |
>
> We have clarified this point more explicitly in the revision.
>
> ---
>
> #### **2. Response to Weakness 2 & Question 2: About the Universality of the "Shallow=General" Assumption and Task-Dependent Pruning Strategies**
>
> We thank the reviewer for this insightful and profound question. We would like to offer the following three points for clarification:
>
> - The "shallow=general, deep=specific" assumption that our method relies on is **built upon** a substantial body of research on the layer-wise functional analysis of deep neural networks, especially Large Language Models.
>   - For instance, research by Tenney et al. (2019) [1] showed that BERT's layers rediscover the classical NLP pipeline (POS tagging -> parsing -> semantic role labeling). Dai et al. (2022) [2] and Belrose et al. (2023) [3] have revealed that the shallower layers of LLMs focus on capturing fundamental linguistic structures like grammar and syntax, while deeper layers integrate this information to form higher-level, more abstract semantic representations.
>
> - We conducted a downstream tasks capability analysis in Figure 3. The results show that our pruning strategy achieves the most significant performance gains on tasks that heavily rely on precise language understanding. This strongly supports our core hypothesis: by conservatively pruning the shallow layers, we can effectively **preserve the model's fundamental language representations**.
>
> - Our work provides preliminary evidence for the assumption that the "optimal pruning strategy is task-dependent." The strategy we currently employ can be seen as a baseline suitable for general downstream tasks. Exploring adaptive mechanisms that can dynamically adjust the pruning rate allocation based on task characteristics is precisely the next step we are actively pursuing.
>
> We have emphasized this more prominently in the revision.
>
> [1] Tenney, Ian, Dipanjan Das, and Ellie Pavlick. "BERT Rediscovers the Classical NLP Pipeline." Proceedings of the 57th Annual Meeting of the Association for Computational Linguistics. 2019.
>
> [2] Dai, Damai, et al. "Knowledge neurons in pretrained transformers." Proceedings of the 60th Annual Meeting of the Association for Computational Linguistics (Volume 1: Long Papers). 2022.
>
> [3] Belrose, Nora, et al. "Eliciting latent predictions from transformers with the tuned lens." arXiv preprint arXiv:2303.08112 (2023).

---

> > ### Comment · Reviewer_MVYf · 2025-11-26
> > **Follow-up Questions Regarding Adaptivity and Layer-wise Representation Assumptions**
> >
> > Thank you very much for your detailed and thoughtful response. While reading your clarifications, I developed two follow-up questions that I believe could lead to an interesting discussion:
> >
> > 1. **On the adaptivity suggested by Table 4:**
> >    In the Appendix, Table 4 shows that under different sparsity ratios, the *optimal* HPRS scheduling strategy is not the same. Would this observation itself suggest that a “more adaptive” or dynamically adjusted scheduling mechanism may be necessary, rather than relying on a predefined family of monotonic schedules?
> >
> > 2. **On the “shallow = general, deep = specific” assumption:**
> >    Recent work on *Loop Transformers* appears to challenge or refine this assumption. Both the training-free formulation [1] and the trainable variant [2] demonstrate that recurrently routing deeper hidden states back into shallower layers yields performance improvements.
> >    Would this imply that feature representations across depth may not follow a strictly hierarchical progression from general (shallow) to specific (deep), but instead exhibit a more intertwined structure—potentially influencing how pruning strategies reason about the functional roles of different layers?
> >
> > I am very interested in your perspective on these points, and I believe they may help further refine or deepen the conceptual understanding of the proposed framework.
> >
> > [1] Skip a Layer or Loop it? Test-Time Depth Adaptation of Pretrained LLMs. https://arxiv.org/abs/2507.07996
> > [2] Scaling Latent Reasoning via Looped Language Models. https://arxiv.org/abs/2510.25741

---

> > > ### Author Response · Authors · 2025-11-29
> > >
> > > We greatly appreciate these two highly insightful follow-up questions. Here are our thoughts and responses:
> > >
> > > **1. About the more adaptive scheduling mechanism suggested by Table 4**
> > >
> > > We strongly agree with your insight. The variations in optimal scheduling strategies across different sparsity ratios in Table 4 indeed suggest that a fully dynamic or learnable scheduling mechanism could potentially outperform a predefined monotonic strategy.
> > >
> > > - However, in our HA-PAT framework, **HPRS was introduced primarily as an inductive bias to regularize the search space.** Since we perform pruning simultaneously with fine-tuning, the optimization landscape is vast and noisy. Without the global guidance provided by HPRS, training often suffers from instability or converges to suboptimal solutions. HPRS provides a stable initial solution space, ensuring training convergence and efficiency.
> > > - It is worth noting that while HPRS defines a global trend, our proposed LIM allows each layer to make local adjustments on top of HPRS. This realizes a form of **constrained adaptivity**.
> > > - As you suggested, exploring more adaptive pruning ratio scheduling mechanisms is indeed a key direction for our future work. We have explicitly acknowledged this in the "Limitations and Future Work" section of the paper.
> > >
> > > **2. Discussion on Loop Transformers and the “Shallow=General” assumption**
> > >
> > > The works on Loop Transformers [1, 2] you mentioned indeed suggest that feature representations may possess a more complex, non-strictly linear intertwined structure. After careful examination, we find that **HA-PAT's design philosophy does not contradict these observations; rather, they complement and corroborate each other on three levels**:
> > >
> > > - We observed that in Section "5.2 Layer Engagement" of CoLa [1], the authors found: `Across all settings, early layers are most frequently engaged, likely due to their broad utility in extracting low-level features.` **This finding strongly supports the "shallow = general" premise in HA-PAT**. It implies that even models allowing layer skipping or looping still rely on shallow layers to process fundamental linguistic features. Therefore, HPRS assigning a lower pruning ratio to shallow layers aligns with the intrinsic properties of these models.
> > >
> > > - We wish to clarify the distinction between HA-PAT and Loop Transformers:
> > >   - Our work focuses on the **standard feed-forward Transformers** (e.g., LLaMA, Gemma) that currently dominate the industry. In these architectures, the depth of information processing is strictly bound to the **physical layer count**. Extensive existing research [3-5] confirms that in a single forward pass, features indeed evolve from "specific" to "abstract" along the physical layers.
> > >   - Loop Transformers, by introducing recurrent routing, collapse the evolution from "general to specific" into different **time steps** over the same set of weights.
> > >   - **Regardless of the architecture, information processing generally evolves from feature extraction to logical reasoning.** HA-PAT exploits this by compressing redundancy in the deep layers. While Loop methods require complex dynamic routing, HA-PAT offers hardware-agnostic acceleration via structural pruning, ensuring efficient deployment on standard hardware.
> > >
> > > - Finally, your insight regarding the "Intertwined Structure" actually further highlights the advantage of HA-PAT over traditional uniform pruning.
> > >   - If feature representations indeed exhibit a highly intertwined structure (e.g., deep semantic features appearing in shallow layers), **traditional uniform pruning strategies would fail even more significantly**, as they cannot distinguish the deep logical components potentially present in shallow layers.
> > >   - While HA-PAT uses HPRS as an initial guide, the core **LIM mechanism is data-driven**. During fine-tuning, if a shallow layer becomes critical because it undertakes complex routing or higher-order reasoning functions (similar to those mentioned in Loop Transformers), the gradients from the task loss will automatically prevent that layer's LIM mask from converging to zero. Therefore, through LIM, HA-PAT possesses the adaptive capability to handle non-uniform or even non-strictly hierarchical feature distributions.
> > >
> > > Thank you again for providing these profound insights and references. We have added a discussion on Loop Transformers to the "Limitations and Future Work" section.
> > >
> > > [1] Skip a layer or loop it? test-time depth adaptation of pretrained llms. https://arxiv.org/abs/2507.07996
> > >
> > > [2] Scaling Latent Reasoning via Looped Language Models. https://arxiv.org/abs/2510.25741
> > >
> > > [3] BERT Rediscovers the Classical NLP Pipeline. https://arxiv.org/abs/1905.05950
> > >
> > > [4] Knowledge neurons in pretrained transformers. https://aclanthology.org/2022.acl-long.581/
> > >
> > > [5] Eliciting latent predictions from transformers with the tuned lens. https://arxiv.org/abs/2303.08112

---

> ### Author Response · Authors · 2025-11-21
> **Response to Reviewer kzTW (2/2)**
>
> #### **3. Response to Question 1: About the Computational Overhead Introduced by LIM**
>
> Compared to the global mask (USM) in PAT, **the computational overhead introduced by LIM is negligible**. This table clearly illustrates the additional overhead introduced by LIM:
>
> | Model | Method | Total Parameters | Trainable Parameters | Trainable Ratio |
> | :---: | :---: | ---: | ---: | :---: |
> | llama2-7B | LoRA only | 6,788,384,816 | 49,971,200 | 0.74% |
> | - | Original PAT | 6,895,153,152 | 156,737,536 | 2.27% |
> | - | PAT+LIM | 6,895,415,296 | 156,999,680 | 2.28% |
> | llama2-13B | LoRA only | 13,094,097,920 | 78,233,600 | 0.60% |
> | - | Original PAT | 13,260,405,760 | 244,541,440 | 1.84% |
> | - | PAT+LIM | 13,260,815,360 | 244,951,040 | 1.85% |
> | Gemma-2B | LoRA only | 2,530,686,976 | 24,514,560 | 0.97% |
> | - | Original PAT | 2,576,230,400 | 70,057,984 | 2.72% |
> | - | PAT+LIM | 2,576,304,128 | 70,131,712 | 2.72% |
> | Gemma-7B | LoRA only | 8,600,185,856 | 62,504,960 | 0.73% |
> | - | Original PAT | 8,705,426,432 | 167,745,536 | 1.93% |
> | - | PAT+LIM | 8,705,598,464 | 167,917,568 | 1.93% |
>
> - The additional trainable parameters of LIM come only from the independent masks for each layer, totaling $2 \times L \times d_{hidden}$ (where $L$ is the number of layers and $d_{hidden}$ is the hidden dimension). This number is extremely small compared to the extra parameters introduced by PAT.
> - For Llama2-7B, adding LIM only increases the trainable parameters by **0.26M**. This increment is merely **0.24%** of the parameter increase from PAT, and it only slightly raises the overall trainable parameter ratio from 2.27% to 2.28%. Furthermore, in our experiments, **the actual training time of HA-PAT was only 5%~10% longer than that of PAT**.
>
> We have added a detailed analysis of the computational overhead to the appendix in the revision.

---

### Author Response · Authors · 2025-11-30
**Summary to Area Chair**

Dear Area Chair,

We sincerely thank you and the reviewers for the time and effort dedicated to reviewing our paper. During the rebuttal period, we have engaged in active discussions with all reviewers. We have provided detailed clarifications, additional theoretical grounding, and new experimental results to address their concerns.

Below is a summary of our exchanges with each reviewer and the key improvements made to the paper.

---

### **Response to Reviewer kzTW**

- **Core Concern:** The universality of HPRS and whether it is truly "adaptive"; the computational overhead of LIM.
- **Our Response:**
  - We clarified that HPRS serves as an inductive bias, while true adaptivity comes from LIM automatically learning intra-layer sparse structures based on gradients.
  - Experiments demonstrate that HA-PAT outperforms the uniform pruning baseline regardless of the schedule function (Linear, Cosine, or Sigmoid), proving the hierarchical principle is more critical than the specific function.
  - Data shows that LIM adds only 0.24% trainable parameters and increases training time by merely 5%-10%, making the overhead negligible.
  - We explained that HA-PAT's design complements findings from Loop Transformers, and HA-PAT's data-driven masks are capable of handling non-strict hierarchical structures.

---

### **Response to Reviewer a3EE**

- **Core Concern:** Lack of technical novelty (claiming LIM/HPRS are common); missing strong baselines; questioning the necessity of ACO.
- **Our Response:**
  - We emphasized that the core contribution of HA-PAT lies in integrating these components into the Pruning-Aware Tuning paradigm for the first time, enabling end-to-end simultaneous optimization, which fundamentally differs from traditional "prune-then-finetune" or "distillation" approaches (e.g., ShearedLLaMA, Minitron).
  - We reproduced and compared against the recent MaGrip method. Results show that HA-PAT significantly outperforms MaGrip on the Gemma-7B model across multiple tasks.
  - New ablation studies (on Gemma models) demonstrate that ACO further boosts performance on top of LIM and HPRS, validating the value of decoupling direction and magnitude.
- **Status**: The reviewer acknowledged our clarification on baselines and ACO, **explicitly stating they  would like to raise their rating**.

---

### **Response to Reviewer qV2M**

- **Core Concern:** Lack of theoretical justification for HPRS; ACO appears identical to the original PAT method; inter-layer feature alignment issues after pruning.
- **Our Response:**
  - We elaborated on the Information Bottleneck (IB) theory and cited recent findings on "identity mapping degeneration" in deep layers to theoretically justify our aggressive pruning in deeper layers.
  - We detailed how ACO explicitly decouples "magnitude" from "direction" by introducing an independent scaling vector $v$. This is a significant improvement over the implicit implementation in the original PAT, and experiments verify its performance gains.
  - We explained how residual connections and the end-to-end training mechanism automatically ensure inter-layer feature alignment.
  - We added experiments on Llama-3.1-8B and Llama-3.2-3B, proving that HPRS remains effective on the latest model architectures.

---

### **Response to Reviewer MVYf**

- **Core Concern:** Learnability of HPRS; scalability to larger models; hyperparameter sensitivity.
- **Our Response:**
  - We acknowledge that fully automated schedule learning is a future direction, but the current predefined schedule provides necessary inductive bias, helping stabilize convergence in a vast search space.
  - While limited by compute resources to test >30B models, the robust performance across 2B, 7B, 8B, and 13B scales demonstrates its potential.
  - We conducted a detailed sensitivity analysis in Appendix A.4, proving that the model is robust to parameters like pruning range $\Delta p$, and performance improves as hierarchical differentiation increases, further validating the hierarchical hypothesis.

---

We hope these clarifications address the reviewer’s concerns and contribute to a positive reassessment of our work.

Best regards,

The Authors

---

### Meta-Review · Area_Chair_GGkx · 2026-01-09

**Summary:**

This paper received 4 reviews. The reviewers (score/confidence) are: `MVYf (4/5), a3EE (4/4), qV2M (4/4), kzTW (6/3)`.

Their major concerns lie in the methodology, experiments, and presentation aspects:

Methodology:
- The Hierarchical Pruning Ratio Scheduling (HPRS) is predefined rather than learnable, limiting adaptability to diverse architectures or tasks `MVYf (4/5)`.
- HA-PAT lacks sufficient technical novelty, as Layer-wise Independent Masks (LIM), HPRS, and Adaptive Compensation Operator (ACO) are all derived from existing LLM pruning methods `a3EE (4/4)`.
- HPRS is presented as an empirical heuristic without formal theoretical justification, weakening confidence in its universality across architectures `qV2M (4/4)`.

Experiments:
- Scalability analysis is missing for models larger than 30B, and there is no exploration of inference-time deployment on real hardware `MVYf (4/5).`
- The paper lacks strong baselines (e.g., Puzzle, one of the SoTA layer-wise adaptive pruning methods) and uses outdated models `a3EE (4/4).`
- It is unclear whether LIM introduces significant computational overhead during the pruning-aware tuning phase compared to the global mask in PAT `kzTW (6/3)`.

Presentation:
- Notations in Section 3.2 are confusing and non-standard (e.g., using uppercase letters for vectors), and the equivalence between Eq. 3 and the forward pass of a pruned model is not explained `a3EE (4/4).`

**Reviewer Concerns:**

The paper has three negative and quite confident reviewers: `MVYf (4/5), a3EE (4/4), qV2M (4/4)`

- a3EE explicitly stated that his/her major concerns have been addressed and will raise the score accordingly.

- qV2M was concerned that the proposed HPRS is empirical and not theoretically justified. The authors respond with IB theory. The reviewer did not reply any further. Based on the discussions, the concern is still outstanding by my best judgment.

- MVYf questioned the limited learnability of HPRS. The authors responded by explaining why this is reasonable (citing the IB principles etc). But this did not change the fact that the HPRS is not learned and potentially leads to the generalization problem to new models. The reviewer also requested results on larger models (such as 30B models). The authors said that this is not feasible due to their limited computational resources. Overall, the authors clarified some issues but many concerns are still outstanding.

In summary, a3EE's concerns are resolved. MVYf and qV2M's concerns are outstanding.

**Reviewer Scores:**

After checking the rebuttal, also based on my research experience in the area,

a3EE will raise the score (the reviewer has explicitly mentioned this);

MVYf and qV2M will probably maintain the score.

This leads to a probable final score/confidence: `MVYf (4/5), a3EE (6/4), qV2M (4/4), kzTW (6/3)`. Considering the potential impact of this paper, along with the other papers in my batch, I believe this paper falls below the acceptance threshold.

---

### Decision · Program_Chairs · 2026-01-26

Reject